# Revisiting Audio-language Pretraining for Learning General-purpose Audio Representation

## Abstract

Audio-language pretraining holds promise for leraning general-purpose audio representation, yet remains underexplored compared to its vision counterpart. Crucially, there is no consensus on whether audio–language models can build effective general-purpose audio encoders, nor a systematic understanding of how pretraining objectives behave across diverse audio processing tasks and scales. We identify three key barriers: limited large-scale audio-text corpora, insufficient caption diversity, and lack of systematic exploration and evaluation. To fill this gap, we present the first principled empirical study of audio–language pretraining. To this end, we introduce CaptionStew, a 10.7M caption dataset aggregating diverse open-source audio-text corpora across multiple domains and captioning styles. Using this resource, we conduct the first comprehensive evaluation comparing contrastive and captioning objectives for audio representation learning across speech, music, and environmental sound tasks. Our results not only demonstrate that audio-language pretraining yields competitive, transferable representations, but also reveal critical trade-offs: contrastive learning offers superior data efficiency, while captioning exhibits better scalability. Furthermore, we find that supervised initialization provides diminishing returns at scale, challenging common practices. By grounding these claims in empirical evidence, we establish a viable pathway toward general-purpose audio representation learning, guiding future research. To accelerate progress, we will release data preparation recipes, training protocols, and pretrained models.

## 1 Introduction

Representation learning has long been central to audio processing[1], with substantial progress over the past decades. Early advances relied on supervised learning, where models trained on labeled corpora were adapted to related downstream tasks or transferred across domains (Kong et al., 2020; Chen et al., 2022a; Snyder et al., 2018; Desplanques et al., 2020). More recently, self-supervised learning (SSL) has emerged as the promising paradigm. By pretraining on large-scale unlabeled audio with contrastive objectives or masked modeling (Gong et al., 2022a; Chen et al., 2023; Baevski et al., 2020; Hsu et al., 2021; Li et al., 2024), the resulting models learn rich structural knowledge of audio signals, consistently enhancing performance across many speech and audio benchmarks (Yang et al., 2021; Turian et al., 2022; Yuan et al., 2023).

While these techniques have achieved remarkable success, a fundamental limitation persists: existing methods are primarily designed to excel on specific tasks. This domain specificity stems from explicit inductive biases embedded in model architectures and training objectives. Models optimized for environmental sounds usually underperform capturing speaker characteristics or paralinguistic information in speech, and vice versa (Turian et al., 2022). Achieving general-purpose audio representations that transfer robustly across diverse audio processing tasks remains a challenging and actively pursued goal in the field.

---

[1]In this work, audio processing refers to audio understanding, speech analysis and music understanding, while excluding automatic speech recognition

An emerging and promising alternative is audio–language pretraining (Elizalde et al., 2023; Wu et al., 2023), which grounds audio perception with natural language descriptions (e.g. captions). In this framework, text serves as a flexible semantic scaffold, offering supervision potentially spanning multiple levels of granularity, from coarse event categories to fine-grained acoustic attributes, offering a offering a unified path toward general audio understanding (Sakshi et al., 2025; Huang et al., 2025; Yang et al., 2024b; Su et al., 2025).

The success of vision–language pretraining underscores this promise. Models like CLIP (Radford et al., 2021) and AIM-v2 (Fini et al., 2025) not only power vision–language tasks but also produce representations that benefit a broad range of vision tasks (Liu et al., 2023; Minderer et al., 2022; Crowson et al., 2022). In contrast, audio–language models have not yet seen similar adoption. Existing models such as CLAP (Elizalde et al., 2023; Wu et al., 2023) are primarily restricted to retrieval tasks. Consequently, the audio community still lacks a systematic understanding of whether audio–language pretraining is viable as a general-purpose representation learning framework. Fundamental questions remain unanswered: How do different pretraining objectives behave or scale? How does transfer performance vary across heterogeneous audio processing tasks like speaker identification versus audio event classification? To our knowledge, the absence of empirical evidence regarding these questions has hindered progress and led to uncertainty and inconsistency of design choices. We identify three key challenges that have constrained progress. First, large-scale, web-mined image–text corpora (Schuhmann et al., 2022; Gadre et al., 2023) contain billions of pairs, but no comparable resource exists for audio. Current audio caption datasets barely exceed one million pairs (Bai et al., 2025; Mei et al., 2024; Kim et al., 2019; Drossos et al., 2020), often relying on captions synthesized or augmented by large language models, fundamentally limiting the scaling potential of audio–language models. Second, widely used audio caption corpora focus predominantly on identifying what is presenting in the audio, while providing limited coverage of the rich range of acoustic attributes that characterize different audio signals. For instance, captions rarely characterize speaker characteristics (voice timbre, speaking style), musical attributes (harmonic structure, rhythmic patterns), or environmental acoustics (reverberation, background ambiance). This imbalanced focus limits the model's ability to learn representations that capture the full range of audio semantics. Third, prior work has primarily focused on contrastive learning (Elizalde et al., 2023; Wu et al., 2023; 2022) and evaluated on audio–text retrieval. Systematic studies on alternative pretraining objectives (e.g., captioning) and comprehensive evaluations across a wide suite of audio understanding tasks remain scarce, limiting our understanding of what drives effective audio–language pretraining.

In this work, we therefore revisit audio–language pretraining with the goal of reestablishing its viability as a pathway toward general-purpose audio representation learning. We do not propose a new method; instead, we aim to provide a foundational empirical study to fill the critical knowledge gap described above, establishing an rigorous baseline to guide future research in accordance with scientific best practices. Specifically, our contributions are:

- We introduce **CaptionStew**, a large-scale aggregation of diverse open-source audio–text datasets spanning multiple domains and captioning styles, addressing the data scarcity and diversity limitations in current audio-language pretraining.

- We provide the first comprehensive evaluation of audio-language pretraining across diverse tasks and protocols, demonstrating that audio–language pretraining produces competitive, transferable representations across speech, music, and environmental audio domains.

- We conduct the first systematic comparison of contrastive learning and captioning objectives for audio representation learning, revealing that contrastive learning exhibits superior data efficiency while captioning demonstrates better scalability.

- We analyze key training factors including data scaling effects and supervised pretraining initialization, showing that while AudioSet pretraining provides general benefits, its effects diminish for tasks unrelated to audio event classification and at larger data scales, challenging common practices in the field.

Our study reveals actionable insights that were previously undocumented in audio literature and occasionally contradict trends from other modalities. Collectively, the results validate audio–language pretraining as a practical and competitive approach for learning general-purpose audio representations. To accelerate progress in this direction, we will release data preperation recipes, training and evaluation scripts, and pretrained models.

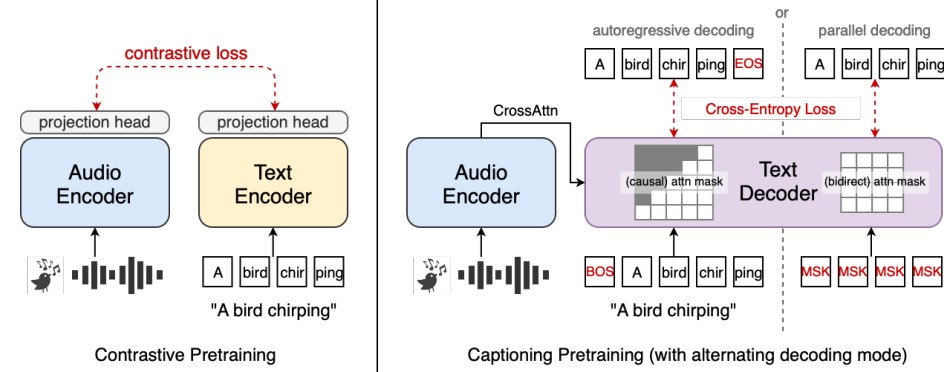

Figure 1: Audio-language pretraining objective studied in this work: contrastive and captioning.

## 2  LANGUAGE-AUDIO PRETRAINING

Audio–language pretraining learns audio representations by establishing correspondence between audio signals and natural language descriptions. The core objective is to leverage text as structured semantic supervision, enabling models to capture diverse information across speech, music, and environmental sounds within a unified framework. Audio–language models typically employ a two-tower architecture: an audio encoder $f_a$ that maps raw audio signals into contextual representations, and a text component $f_t$ whose design depends on the training objective. As shown in Figure 1, we explore two complementary paradigms that differ fundamentally in how they establish audio-text correspondence, contrastive and captioning objective. These approaches represent discriminative and generative perspectives on audio-language alignment, respectively.

**Contrastive Objective** is proven to be a robust representation learning method (Chen et al., 2020b; Radford et al., 2021; Baevski et al., 2020) and have been a dominant approach for audio-language pretraining (Elizalde et al., 2023; Wu et al., 2023; 2022). This approach aligns audio and text representations in a shared embedding space by maximizing similarity between paired samples while minimizing similarity between mismatched pairs. Given a batch of paired samples $\{(a_i, t_i)\}_{i=1}^N$, the audio encoder produces frame- (or patch-) level representations that are pooled and projected to audio embeddings $\mathbf{z}_i^a$, while the text encoder $f_t$ generates corresponding text embeddings $\mathbf{z}_i^t$. The symmetric InfoNCE loss (Oord et al., 2018) is applied to optimize both modalities:

$$\mathcal{L}_{\text{con}} = -\frac{1}{2N} \sum_{i=1}^N \left[ \log \frac{\exp(\text{sim}(\mathbf{z}_i^a, \mathbf{z}_i^t)/\tau)}{\sum_{j=1}^N \exp(\text{sim}(\mathbf{z}_i^a, \mathbf{z}_j^a)/\tau)} + \log \frac{\exp(\text{sim}(\mathbf{z}_i^t, \mathbf{z}_i^a)/\tau)}{\sum_{j=1}^N \exp(\text{sim}(\mathbf{z}_i^t, \mathbf{z}_j^a)/\tau)} \right], \quad (1)$$

where $\text{sim}(\cdot, \cdot)$ denotes cosine similarity and $\tau$ is a learnable temperature parameter. This objective encourages paired audio-text samples to be close in embedding space, encouraging semantic organization where similar content is grouped together.

**Captioning Objective** takes a generative approach to audio-language alignment, learning representations by generating textual descriptions from audio. We argue that captioning presents a a promising yet underexplored alternative for audio–language pretraining. Theoretically, the cross-attention mechanism provides frame-level supervision on the audio representation, offering denser and more structured learning signals than the utterance-level alignment used in contrastive learning. Also, because captioning models the joint distribution over all caption tokens, it is inherently more sensitive to fine-grained attributes, relations, and word order, enabling richer relational grounding Yuksekgonul et al. (2023); Hsieh et al. (2023); Tschannen et al. (2023). Moreover, caption-based supervision is increasingly relevant given recent efforts toward general audio understanding systems (Dinkel et al., 2025; Goel et al., 2025)

Given an audio signal $a_i$, the encoder $f_a$ produces contextual representations $\mathbf{Z}_i^a$, which are fed into a transformer decoder $g_t$ through cross-attention. Inspired by CapPa (Tschannen et al., 2023), we alternate between two decoding modes—autoregressive and parallel prediction—to enhance audio encoder representation learning. In the autoregressive decoding, the decoder generates caption

tokens $(y_1, \ldots, y_T)$ sequentially, with each token conditioned on the audio representation and previously generated tokens. Training follows the teacher-forcing approach with a cross-entropy loss:

$$\mathcal{L}_{\text{cap}} = -\sum_{t=1}^{T} \log p_\theta(y_t \mid y_{<t}, \mathbf{Z}_i^a), \tag{2}$$

In parallel prediction, we replace the decoder input tokens with [MASK] tokens and remove the causal attention mask, forcing simultaneous prediction of all tokens based solely on audio features:

$$\mathcal{L}_{\text{par}} = -\sum_{t=1}^{T} \log p_\theta(y_t \mid \mathbf{Z}_i^a), \tag{3}$$

This mode eliminates reliance on prior autoregressive context and forces each token prediction to depend solely on the audio representation, thereby strengthening encoder supervision. In a preliminary experiment, we observe that incorporating the parallel mode yields stronger representations than using a purely autoregressive decoder. We adopt mixed training where a random fraction of each minibatch uses standard autoregression while the remainder use parallel decoding.

## 3 CAPTIONSTEW DATASET

To investigate the potential of audio–language pretraining for general-purpose representation learning, we collect a large-scale and diverse audio caption dataset that addresses key limitations in existing corpora. Audio signals inherently encode information across multiple dimensions—timbre, pitch, rhythm, semantic events, emotional tone, and acoustic environment—each amenable to different linguistic descriptions. However, existing large-scale audio caption datasets typically rely on a single caption-generation pipeline (Appendix A.2), where all captions are produced through the same procedure—either human annotation following uniform guidelines or LLM-based synthesis—and consequently share a homogeneous linguistic style. This uniformity offers consistency and scalability but introduces systematic stylistic biases and restricts linguistic diversity. Moreover, single-pipeline captions tend to exhibit limited syntactic variation and a narrow descriptive focus on only a subset of audio characteristics, often overlooking complementary acoustic attributes.

To fully leverage text as a flexible semantic scaffold for diverse audio representation learning, we embrace caption diversity across sources, styles, and descriptive granularities. Rather than creating captions through a single pipeline, we aggregate existing open-source corpora (Kim et al., 2019; Drossos et al., 2020; Agostinelli et al., 2023; Mei et al., 2024; Chen et al., 2025; Bai et al., 2025; Diwan et al., 2025; Roy et al., 2025). These datasets span multiple audio domains—general sound events, expressive speech, and musical performance—and employ fundamentally different caption creation methodologies. This aggregation yields captions that describe complementary audio aspects with varying granularity, from coarse event categories to fine-grained acoustic attributes. Please refer to Appendix A.2 for detail and examples of each source dataset. When multiple datasets contain identical audio samples with different captions, we identify these overlaps and consolidate all available captions for each audio file. This multi-caption pairing allows single audio clips to benefit from diverse linguistic variation and descriptive focuses, enriching the supervision signal. To ensure evaluation integrity, we carefully filter out samples overlapping with development or test sets of downstream benchmarks.

The resulting dataset, **CaptionStew** (denoted by CS10M), contains 9.3 million audio samples paired with 10.7 million captions, spanning 37,290 hours across speech, music, and environmental domains. Compared to existing collections, CaptionStew achieves both greater scale and broader coverage. This not only facilitates the learning of general-purpose audio representations but also provides a standardized, reproducible testbed for rigorous empirical study. Table 1 presents a comparison with existing audio caption datasets.

Table 1: Comparison of publicly available audio caption datasets. The number of audio-text pairs (#pair) and number of unique words (#vocab) are shown here.

| Audio Caption Dataset | #pair | #vocab |
|---|---|---|
| *Human-annotated* | | |
| AudioCaps Kim et al. (2019) | 46K | 4,844 |
| Clotho Drossos et al. (2020) | 5K | 4,366 |
| MusicCaps Agostinelli et al. (2023) | 5K | 3,730 |
| *LLM-augmented* | | |
| WavCaps Mei et al. (2024) | 403K | 18,372 |
| AudioSetCaps Bai et al. (2025) | 1.9M | 21,783 |
| FusionAudio Chen et al. (2025) | 1.2M | 18,403 |
| AutoACD Sun et al. | 1.5M | 20,491 |
| CaptionStew (Ours) | 10.7M | 56,586 |

Table 2: Datasets used for evaluating linear probing, audio-language task and open-form question answering performance (separated by lines). All metrics are higher the better. [†]reported with AIR-Bench Yang et al. (2024b).

| Evaluation Dataset | Task | Metrics |
|---|---|---|
| FSD-50k | Multi-label audio event classification | mAP |
| VggSound | Single-label audio event classification | accuracy |
| VoxCeleb2 | Speaker identification | accuracy |
| CREMA | Speech emotion recognition | accuracy |
| MagnaTagATune | Music tagging | mAP |
| NSynth | Musical instrument classification | accuracy |
| AS-strong | Sound event detection | PSDS1 |
| AudioCaps | Text-to-audio retrieval | Recall@1 |
| ParaSpeechCaps | Audio captioning | RougeL |
| MusicCaps | | |
| ClothoAQA | | |
| ParaLMQA | Open-formed question answering | Score[†] |
| MusicQA | | |

## 4 EXPERIMENTAL SETUP

### 4.1 IMPLEMENTATION DETAILS

We pretrain all models on CaptionStew. All audio is resampled to 16 kHz and converted into 80-dimensional log-Mel filterbank features using a 25 ms window length and 10 ms hop size. Text is tokenized with a 50k-vocabulary BPE tokenizer (Lewis et al., 2020).

The audio encoder uses a Zipformer-M architecture (Yao et al., 2024), chosen for its efficiency on long sequences and fast convergence. Zipformer employs six encoder blocks in a U-Net structure that processes sequences at multiple resolutions to capture fine- and coarse-grained temporal information. Although originally designed for automatic speech recognition, our preliminary experiments confirm Zipformer as a competitive backbone across audio classification tasks (see Appendix A.3). For contrastive pretraining, the text encoder follows BERT-base architecture (12 layers 768 hidden dimensions) (Devlin et al., 2019). For captioning pretraining, the text decoder adopts the BART-base decoder architecture (6 layers, 768 hidden dimensions) (Lewis et al., 2020). We use twice as many encoder layers as decoder layers to ensure comparable training speed across objectives.

Following prior works in audio-language pretraining (Elizalde et al., 2023; Wu et al., 2023; Mei et al., 2024; Bai et al., 2025), we experiment with two scenarios: training from scratch (denoted by *-scratch*) or initialized from pretrained checkpoints (denoted by *-init*). The audio encoder initializes from a Zipformer-based audio event classifier trained on AudioSet (Gemmeke et al., 2017) with an mAP of 0.46, while text components use corresponding publicly available checkpoints. All models are trained on 8 Tesla V100 GPUs with an effective batch size of 640 seconds of audio per GPU. Training runs for 600k steps from scratch (14 days wall-clock time) or 200k steps if initialized from pretrained checkpoint.

### 4.2 EVALUATION PROTOCOLS AND DATASETS

We evaluate pretrained audio encoders across three protocols assessing discriminative capabilities, audio-language alignment, and open-formed question answering. All experiments probe frozen representations from the audio encoder's final layer to ensure fair model comparison. Table 2 and Appendix A.4 details the datasets and metrics for each task.

**Linear Probing** trains simple linear classifier on frozen representations. For detection tasks, we adopt the frame-level representation as the input for the linear head. For classification tasks, we experiment with two pooling mechanisms—mean pooling and multi-head attention pooling (Lee et al., 2019)—to aggregate frame-level features into clip-level embeddings before feeding them into the linear head. We evaluate across a diverse set of tasks across audio domains, including audio event classification (AEC) (Fonseca et al., 2021; Chen et al., 2020a), sound event detection (SED) (Her-

shey et al., 2021), speaker identification (SID) (Chung et al., 2018), speech emotion recognition (SER) (Cao et al., 2014), music tagging (MTAG) (Law et al., 2010) and musical instrument classification (INST) (Engel et al., 2017).

**Audio-language Alignments** follow the LiT protocol (Zhai et al., 2022), adapting pretrained text components to align with frozen audio representations. For retrieval, we pair audio encoders with pretrained RoBERTa-base text encoder (Liu et al., 2019). For captioning, we use pretrained BART-base decoders (Lewis et al., 2020), and only finetune cross-attention layers as we observed more stable training. We evaluate both tasks on a diverse collection of audio-caption datasets spanning multiple audio domains and descriptive focuses: AudioCaps (AC) (Kim et al., 2019) for general sound event descriptions; ParaSpeechCaps (PSC) (Diwan et al., 2025) for speaking-style and acoustic-environment descriptions; and MusicCaps (MC) (Agostinelli et al., 2023) for fine-grained musical attribute descriptions. In all cases, the text-side components are finetuned on the corresponding datasets (Kim et al., 2019; Diwan et al., 2025; Agostinelli et al., 2023), while the audio encoder remains frozen.

**Open-formed Question Answering**. Acknowledging the trend of combining audio encoders with large language models (LLMs) for general audio understanding (Ghosh et al., 2024; Gong et al., 2024), we connects frozen audio encoders to a LLM (Qwen2.5-7B-Instruct Yang et al. (2024a)) through lightweight adaptors that project audio representations into the LLM's embedding space. We train only the adaptor on multiple audio QA datasets that span distinct domains: sound event understanding (Lipping et al., 2022), speaker-related and paralinguistic understanding (Huo et al., 2025), and music understanding (Liu et al., 2024). Evaluation is conducted on the corresponding tracks (sound, speaker-related, music; see Appendix A.4) of AIR-Bench (Yang et al., 2024b). During training, we carefully monitor instruction-following behavior (>99%) to ensure reliable evaluation.

### 4.3 BASELINE METHODS

Recognizing the broad adoption and effectiveness of pretrained audio event classifiers in transfer learning (Alonso-Jiménez et al., 2023; Cappellazzo et al., 2024), audio-language modeling (Elizalde et al., 2023; Wu et al., 2023) and general audio understanding (Gong et al., 2024; Ghosh et al., 2024; Dinkel et al., 2025), we select our pretrained Zipformer-based audio event classifier (denoted by Zipformer-AEC, described in Sec. 4.1) as the primary baseline. In addition, we compare against representative self-supervised learning (SSL) models, each pretrained under different paradigms and specialized for particular audio domains. BEATs (Chen et al., 2023) is an audio SSL model trained with an iterative masked acoustic token prediction framework. Wav2vec 2.0 (Baevski et al., 2020) learns speech representation by distinguishing target quantized latent representations from disctrators. MERT (Li et al., 2024) is a music SSL model trained with masked acoustic modeling, learning to capture acoustic cues and structural information of music. Together, these baselines provide a broad comparative context for studying audio–language pretraining toward general-purpose audio representation.

### 4.4 MAIN RESULTS

We present our evaluation results in Table 3. Our analysis reveals key insights about objective design, representation quality, and the role of initialization.

**Contrastive vs. Captioning Objectives.** The two pretraining paradigms exhibit complementary strengths across evaluation protocols. On linear probing tasks, contrastive learning consistently outperforms captioning, particularly excelling at audio event classification and speaker identification. However, it is worth noting that this gap narrows substantially when the classifier learns to aggregate information across frames through multi-head attention pooling (Appendix A.5). This observation reflects the objectives' inherent designs: contrastive learning explicitly optimizes for linearly separable clip-level representations, while captioning relies on cross-attention mechanisms over frame-level representations for text sequence generation. This finding aligns with recent work highlighting how downstream module choices significantly impact the assessment of audio representation quality (Zaiem et al., 2023). For language-involved tasks, both objectives demonstrate competitive performance, with captioning showing slight advantages in open-form question answering across multiple domains. This suggests captioning's potential for language-involved audio understanding tasks, aligning with recent trends toward generative audio understanding systems.

Table 3: Evaluation results across tasks and protocols. [†]numbers quoted from other papers with consistent evaluation setup. [‡]state-of-the-art results on each task without any training constraints (e.g. full-finetuning) (see Appendix A.5). [††]no available prior work. [‡‡]results of speaker emotion recognition, gender recognition, and age prediction in AIR-Bench Yang et al. (2024b), respectively.

(a) Linear Probing (with mean pooling)

| Method | Model Initialization | Audio-lang. Pretraining | linear probing | | | | | | |
|---|---|---|---|---|---|---|---|---|---|
| | | | AEC FSD50k | AEC VggSound | SID VoxCeleb2 | SER CREMA | MTAG MagnaTagATune | INST NSynth | SED AS-Strong |
| *Existing SSL Models* | | | | | | | | | |
| BEATs Chen et al. (2023) | SSL | – | $0.565^†$ | – | – | – | $0.400^†$ | $\underline{75.90}^†$ | $0.034^†$ |
| Wav2vec 2.0 Baevski et al. (2020) | SSL | – | $0.342^†$ | – | $\underline{51.60}$ | 56.10 | $0.317^†$ | $40.20^†$ | – |
| MERT Li et al. (2024) | SSL | – | – | – | – | – | $0.402^†$ | $72.60^†$ | – |
| *Our Supervised Baselines* | | | | | | | | | |
| Zipformer-AEC Yao et al. (2024) | AudioSet SL | – | 0.656 | $\underline{56.46}$ | 18.84 | 67.14 | 0.407 | 67.19 | $\underline{0.216}$ |
| *Our Audio-lang. Pretrained* | | | | | | | | | |
| Contrastive-*scratch* | – | CS10M | 0.625 | 50.87 | **46.67** | 67.71 | 0.406 | 67.30 | 0.132 |
| Captioning-*scratch* | – | CS10M | 0.580 | 47.79 | 33.43 | 63.60 | 0.401 | 63.10 | 0.124 |
| Contrastive-*init* | AudioSet SL | CS10M | **0.664** | 54.70 | 38.17 | **68.84** | 0.406 | **69.38** | **0.187** |
| Captioning-*init* | AudioSet SL | CS10M | 0.652 | 53.13 | 26.23 | 65.86 | **0.410** | 67.16 | 0.145 |
| SOTA[‡] | | | 0.655 | 59.50 | 96.20 | $-^{††}$ | 0.414 | 79.20 | 0.374 |

(b) Audio-language Alignment / Open-form QA

| Method | Captioning | | | Retrieval | | | Open-formed QA | | |
|---|---|---|---|---|---|---|---|---|---|
| | AC | PSC | MC | AC | PSC | MC | Sound | Speaker-related[‡‡] | Music |
| *Our Supervised Baselines* | | | | | | | | | |
| Zipformer-AEC Yao et al. (2024) | 46.7 | 45.5 | **22.9** | 40.5 | 49.2 | 24.6 | 7.01 | 36.5 / 46.2 / 37.2 | 5.61 |
| *Our Audio-lang. Pretrained* | | | | | | | | | |
| Contrastive-*scratch* | 46.6 | 46.3 | 22.1 | 39.3 | **63.2** | 27.4 | 6.65 | 37.9 / **81.3** / 63.4 | 5.86 |
| Captioning-*scratch* | 46.7 | **46.5** | **22.9** | 36.9 | 60.2 | 23.0 | 6.69 | **44.2** / 65.4 / **69.0** | **5.97** |
| Contrastive-*init* | **47.2** | 46.2 | 22.5 | **42.8** | 60.6 | **29.4** | 6.73 | 35.1 / 67.3 / 64.5 | 5.63 |
| Captioning-*init* | **47.2** | 45.9 | 22.6 | 42.2 | 55 | 28.2 | **7.06** | 32.4 / 49.5 / 45.6 | 5.50 |
| SOTA[‡] | 52.2 | $-^{††}$ | 26.2 | 44.4 | $-^{††}$ | $-^{††}$ | 6.99 | 60.0 / 82.5 / 62.4 | 6.79 |

**Impact of Supervised Initialization.** Initializing from supervised pretraining (AS SL) provides substantial benefits across most tasks, with notable improvements on audio event classification, sound event detection and audio-text retreival. The gains are particularly pronounced for contrastive objectives, suggesting that discriminative pretraining provides useful inductive biases for contrastive learning. However, these benefits diminish (or disappear entirely) when the attributes required for downstream tasks diverge from AudioSet's ontology. On speaker identification and music tagging, scratch-trained models often match or exceed initialized variants, indicating that AudioSet's focus on distinguishing between sound categories may bias representations toward event-level semantics rather than the acoustic attributes (voice timbre, speaking style) or musical structure (genre, harmony, rhythm) essential for these tasks. These findings challenge common initialization practices for audio-language pretraining and suggest the need for tailored pretraining strategies when targeting general-purpose audio representation learning.

**Competitive Performance Across Domains.** Our audio-language representations achieve strong transferability across diverse audio domains. Compared to supervised baselines (Zipformer-AEC), our overall best-performing model (Contrastive-init) demonstrate superior performance on speaker identification, music understanding and audio-text retrieval while maintaining competitiveness on audio-event classification. Against domain-specialized SSL methods (BEATs, Wav2vec 2.0, MERT), our approach consistently shows competitive performance. This consistent cross-domain performance validates our hypothesis that diverse caption aggregation enables broadly transferable representations, establishing audio-language pretraining as a viable path toward learning general-purpose audio representation.

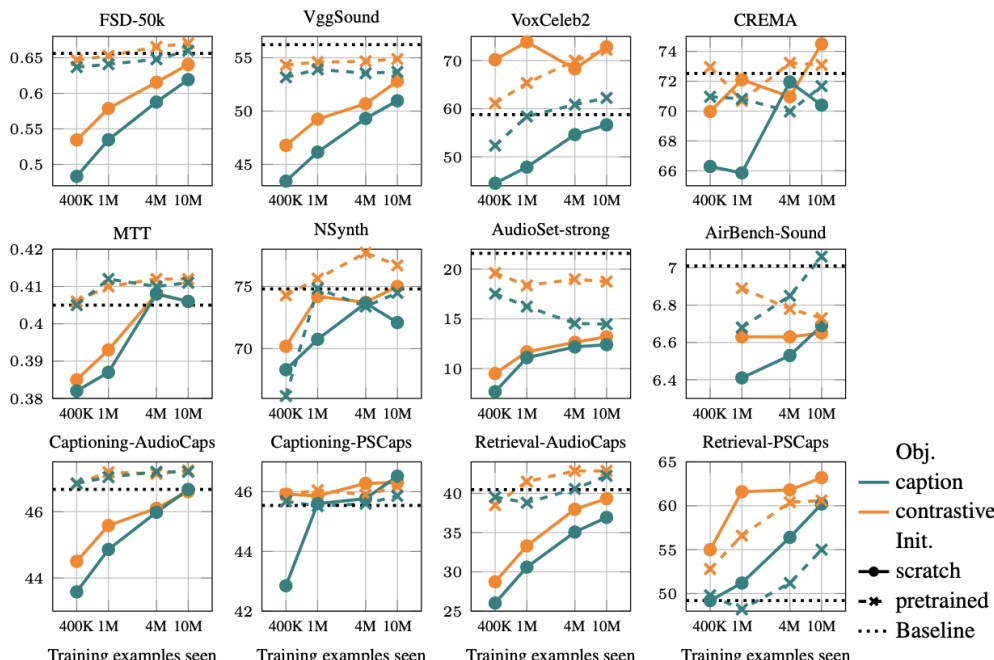

Figure 2: Data scaling behavior of contrastive vs. captioning objectives across representative tasks.

## 4.5 DATA-SCALING EXPERIMENTS

To understand the scalability of audio–language pretraining, we conduct controlled experiments using CaptionStew subsets at 400K, 1M, 4M, and 10M (whole corpus) audio-text pairs. Figure 2 reveals distinct scaling patterns across objectives and evaluation protocols.

**Scaling Patterns.** Most tasks demonstrate consistent performance improvements with increased data scale, validating the potential of large-scale audio-language pretraining. However, notable exceptions emerge that reveal fundamental limitations of current approaches. Sound event detection, particularly for models initialized with AudioSet pretraining, exhibits a reverse scaling trend where performance degrades with more caption data. This suggests a potential conflict between natural language supervision–which typically describes audio characteristics and attributes–and temporal localization tasks requiring precise event boundaries. Additionally, emotion recognition and instrument classification show weaker scaling gains compared to other tasks, likely reflecting limited caption diversity for these specific attributes in existing corpora, which we will discussed in Sec. 4.6.

**Contrastive vs. Captioning Scaling.** Contrastive learning consistently outperforms captioning at varying data scales, particularly under less data and on discriminative tasks such as audio event classification. However, captioning demonstrates slightly better scaling properties, with distinct patterns emerging across task categories. or language-involved tasks–especially captioning and question answering–captioning matches or surpasses contrastive learning at our current 10M-pair scale. On linear probing benchmarks, the gap remains substantial, with scaling trends suggesting captioning would require hundreds of millions of pairs to achieve parity with contrastive methods.

**Impact of Initialization at Scale.** AudioSet initialization provides immediate performance gains but introduces diminishing returns at larger scales. Both contrastive learning and captioning show decreasing benefits from initialization as data scale increases, with scratch and initialized models achieving matched performance at larger scales on some tasks. This suggests that pretrained initialization effectively bootstraps learning at small scales but may constrain the model's ability to adapt to the broader semantic space covered by large-scale caption data, potentially due to mismatch between AudioSet's ontology and diverse audio descriptions.

Overall, these findings reveal complementary behaviors: contrastive pretraining achieves superior data efficiency at current scales, while captioning shows better scalability, especially for language-

Figure 3: t-SNE visualization of sentence embedding of captions grouped by source.

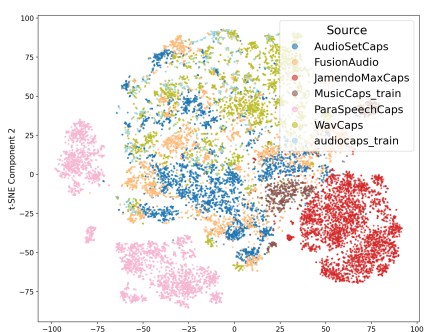

Table 4: Comparison of lexical statistics and diversity across audio caption datasets and text corpora. We report vocabulary size (#vocab), average sentence length (avg. sent), and Distinct-n.

| Source | #vocab | avg. sent | Distinct-n | | | |
|---|---|---|---|---|---|---|
| | | | 1 | 2 | 3 | 4 |
| AudioCaps | 5,572 | 8.46 | 0.011 | 0.113 | 0.309 | 0.519 |
| WavCaps | 18,372 | 7.77 | 0.026 | 0.184 | 0.420 | 0.646 |
| AudioSetCaps | 21,061 | 28.22 | 0.006 | 0.082 | 0.249 | 0.450 |
| FusionAudio | 18,403 | 13.81 | 0.009 | 0.111 | 0.322 | 0.546 |
| JamendoMaxCaps | 27,906 | 63.29 | 0.002 | 0.026 | 0.079 | 0.153 |
| ParaSpeechCaps | 4,060 | 28.50 | 0.001 | 0.015 | 0.051 | 0.112 |
| CaptionStew(Ours) | 56,586 | 32.23 | 0.006 | 0.080 | 0.231 | 0.401 |
| CC12M | 366,175 | 17.03 | 0.046 | 0.486 | 0.813 | 0.927 |
| WikiText-103 | 531,346 | 74.29 | 0.031 | 0.365 | 0.757 | 0.930 |

involved tasks. Importantly, the diminishing returns of initialization at scale indicate that large-scale caption data can provide sufficient semantic supervision independent of domain-specific pretraining, challenging current practices of audio-language pretraining and opening possibilities for learning general-purpose representations from diverse text descriptions alone.

### 4.6 DATASET ANALYSIS

To understand the linguistic characteristics of CaptionStew, we analyze caption diversity across constituent datasets through visualization and quantitative methods. Figure 3 provides compelling evidence of our aggregation strategy's success through t-SNE visualization (Maaten & Hinton, 2008) of sentence embeddings (Reimers & Gurevych, 2019) from sampled captions, revealing distinct clustering patterns by source that demonstrate complementary linguistic perspectives: AudioSet-Caps and WavCaps overlap in audio event descriptions and aligns more with human annotated dataset, while JamendoMaxCaps creates a distinct cluster focused on music-specific terminology, and ParaSpeechCaps forms a separate cluster emphasizing speaking styles and paralinguistic attributes. These minimal overlaps confirm that each dataset contributes distinct caption styles and descriptive focuses.

Quantitative analysis reveals both the benefits and limitations (Table 4). CaptionStew achieves substantial vocabulary expansion (56,586 unique words vs. 4,060-27,906 for individual datasets) However, this growth doesn't yield proportional lexical diversity. CaptionStew's Distinct-n metrics (Li et al., 2015) remain low, falling short of image caption dataset (Changpinyo et al., 2021) and text corpora (Merity et al., 2016). This constraint stems from datasets with limited linguistic variation, particularly JamendoMaxCaps and ParaSpeechCaps with extremely low Distinct-n scores.

These findings highlight that simply combining datasets doesn't guarantee improved linguistic diversity, revealing broader limitations in current audio-language pretraining approaches. Also, the constrained diversity in certain aspect may partially explain weaker scaling behavior observed for certain tasks, as models encounter repetitive linguistic patterns despite increased data volume, aligning with vision-language findings on caption diversity's importance for representation quality (Santurkar et al., 2023; Chan et al., 2022). This analysis motivates developing enhanced aggregation pipeline and more diverse caption generation methods to better capture the full spectrum of information in audio signals, thereby fully realizing the potential of large-scale audio-language pretraining.

## 5 RELATED WORKS

**Audio Representation Learning.** The ultimate goal of audio representation learning is developing a single model suitable for diverse audio understanding tasks. Supervised models trained on labeled datasets have been fundamental to the field, including audio event classifiers (Hershey et al., 2017; Cramer et al., 2019; Kong et al., 2020; Gong et al., 2021; Chen et al., 2022a; Dinkel et al., 2024), speech recognition systems (Radford et al., 2023) and speaker recognition models (Snyder et al., 2018; Desplanques et al., 2020). These approaches remain widely adopted due to their strong

performance on target tasks. In parallel, self-supervised learning methods have emerged as a complementary approach, offering advances across speech (Baevski et al., 2020; Hsu et al., 2021; Chen et al., 2022b; Baevski et al., 2022), audio (Gong et al., 2022a; Huang et al., 2022; Chen et al., 2023; Li & Li, 2022), and music (Li et al., 2024; Zhu et al., 2025) without requiring labeled data. While these methods show improved generalization within their target domains, achieving truly general-purpose audio representations remains challenging.

**Audio–Language Pretraining.** Audio-language models have emerged as a promising approach for learning cross-modal representations. Most existing work focuses on contrastive learning objectives that align audio and text in shared embedding spaces (Elizalde et al., 2023; Wu et al., 2023; 2022). Recent extensions have explored combinations with other objectives (Xu et al., 2023; Zhu et al., 2024; Niizumi et al., 2024; 2025). The field has also witnessed rapid evolution in datasets, transitioning from traditional human-annotated corpora (Kim et al., 2019; Drossos et al., 2020; Agostinelli et al., 2023) to recently constructed LLM-augmented collections (Mei et al., 2024; Bai et al., 2025; Chen et al., 2025; Sun et al.) and domain-specific resources covering speech characteristics (Diwan et al., 2025), and musical attributes (Roy et al., 2025). Our work contributes by providing the first systematic comparison between contrastive and captioning objectives, along with comprehensive evaluation toward general-purpose audio representation.

**Universal Audio Understanding.** The evaluation of audio understanding has evolved from task-specific classification benchmarks (Yang et al., 2021; Turian et al., 2022; Yuan et al., 2023) toward more comprehensive assessment frameworks. Recent developments have emphasized LLM-based audio understanding systems (Ghosh et al., 2024; Gong et al., 2024; Dinkel et al., 2025; Goel et al., 2025; Chu et al., 2024; Tang et al., 2024) that can handle open-form queries and complex reasoning tasks. This shift has driven the development of corresponding evaluation benchmarks that assess models' abilities across diverse audio understanding scenarios, including question answering, reasoning, and multi-step audio analysis (Sakshi et al., 2025; Yang et al., 2024b; Huang et al., 2025; Ma et al., 2025). Our work contributes to this trend by providing the first comprehensive evaluation of audio-language pretraining across discriminative tasks, audio-language alignment, and open-form question answering, thereby bridging the gap between traditional representation learning evaluation and modern universal audio understanding.

## 6  CONCLUSION

We revisited audio–language pretraining with the goal of establishing a rigorous baseline for general-purpose audio representation learning. By aggregating and harmonizing diverse datasets into CaptionStew, we addressed the data scarcity issues that have hindered the field and enabled a rigorous comparison of training objectives and data scales. Our comprehensive evaluation yielded several actionable insights: (1) audio–language pretraining produces competitive representations across speech, music, and environmental sounds; (2) contrastive and captioning objectives exhibit complementary strengths regarding efficiency and scalability; and (3) standard supervised initializations may be unnecessary or even detrimental at scale. Finally, our analysis highlighted the restrictive lexical diversity in current datasets as a key frontier for future improvement. We hope these empirical foundations will accelerate the development of future general-purpose audio representation learning.

## REPRODUCIBILITY STATEMENT

To ensure reproducibility, we provide comprehensive complete source code in the supplementary material. The code includes environmental configuration, training scripts, evaluation protocols, detailed hyperparameter setup and other relevant materials. All experimental components—from model training to evaluation—can be reproduced with runnable scripts in the provided code. We discuss the experimental and evaluation setup in Section 4.1 and Section 4.2.

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

# A APPENDIX

## A.1 LIMITATIONS

While this work provides valuable empirical insights for audio-language pretraining, we acknowledge several important limitations that present opportunities for future research.

**Dataset Construction and Quality.** CaptionStew aggregates captions from multiple sources with varying generation methodologies, including LLM-synthesized descriptions that may introduce systematic biases or artifacts. We do not perform extensive quality control or human verification across the aggregated corpus, which could impact model training. Additionally, our dataset analysis reveals that simple aggregation does not guarantee improved linguistic diversity—CaptionStew's lexical diversity metrics remain lower than mature image-text corpora. However, our design choice prioritizes semantic diversity over linguistic variety, as evidenced by the t-SNE clustering analysis showing distinct descriptive focuses across constituent datasets. While more sophisticated curation strategies could improve quality, our goal was to establish whether diverse caption aggregation can benefit audio representation learning, which our results support despite these limitations.

**Limited Technical Novelty.** Our work primarily combines existing techniques—contrastive learning, captioning objectives, and dataset aggregation—rather than introducing fundamentally new methods. The mixed autoregressive/parallel training approach is adapted from vision-language work (CapPa), and our architectural choices follow standard practices. We acknowledge that the technical contributions are largely empirical rather than methodological. However, this aligns with our primary goal of systematically evaluating audio-language pretraining's potential for general-purpose representation learning. The field currently lacks comprehensive comparative studies across objectives, evaluation protocols, and training factors. Our systematic analysis reveals important insights about scaling behaviors and initialization effects that have practical implications for practitioners, even if the underlying techniques are not novel.

**Limited Model and Data Scalability.** Our experiments are constrained to 10M audio-text pairs and relatively modest model sizes compared to state-of-the-art vision-language systems that leverage billions of samples and much larger architectures. This scale limitation may not fully reflect the potential of audio-language pretraining, particularly for the captioning objective which our results suggest benefits from larger-scale training. Additionally, we do not explore recent advances in large language model integration or more sophisticated architectural designs that could improve performance. These constraints stem from computational resource limitations and our focus on controlled comparisons rather than pushing absolute performance boundaries. Future work with larger scales may reveal different scaling dynamics and stronger evidence for general-purpose capabilities.

Table 5: Details of public-available datasets contribute to proposed CaptionStew dataset. We summarize their size, domain coverage, audio sources, captioning style, and generation pipelines.

| Dataset | #audio/#cap | Domain | Audio source | Caption style | Caption generation pipeline |
|---|---|---|---|---|---|
| AudioCaps (Kim et al., 2019) | 46k/46k | general (environmental, human/animal sounds) | AudioSet (Gemmeke et al., 2017) | Human-annotated, short description | crowdsourced |
| Clotho (Drossos et al., 2020) | 5k/25k | environmental sounds | FreeSound | Human-annotated, short description | crowdsourced |
| MusicCaps (Agostinelli et al., 2023) | 3k/3k | music | AudioSet (Gemmeke et al., 2017) | Expert musician-written, multi-sentence, fine-grained description | expert curation |
| WavCaps (Mei et al., 2024) | 400k/400k | general (environmental, human/animal sounds) | AudioSet (Gemmeke et al., 2017) BBC Sound Effect FreeSound SoundBible | LLM-refined captions | three-stage pipeline: web-crawled raw descriptions → ChatGPT rewrite → filtering |
| AudioSetCaps (Bai et al., 2025) | 1.9M/1.9M 4.0M/4.0M 182k/182k | general (environmental, human/animal sounds) | AudioSet (Gemmeke et al., 2017) YouTube8M (Abu-El-Haija et al., 2016) VggSound (Chen et al., 2020a) | LLM-generated, detailed, multi-sentence description | three-stage pipeline: LALM attribute extraction → LLM captioning → CLAP-based filtering |
| FusionAudio (Chen et al., 2025) | 1.2M/1.2M | general (environmental, human/animal sounds) | AudioSet (Gemmeke et al., 2017) | LLM-augmented, multi-sentence, visual-enhanced description | multimodal context fusion (audio, visual, metadata) + LLM captioning |
| JamendoMaxCap (Roy et al., 2025) | 360k/1.8M | music | Jamendo Platform | LLM-augmented, multi-sentence, fine-grained music description | retrieval-based metadata imputation + LLM captioning |
| ParaSpeechCaps (Diwan et al., 2025) | 116k/116k (base) 924k/924k (scaled) | expressive speech | VoxCeleb1 (Nagrani et al., 2020) VoxCeleb2 (Chung et al., 2018) EARS (Richter et al., 2024) Expresso (Nguyen et al., 2023) Emilia (He et al., 2024) | Human-annotated/LLM-augmented, speaking-style description | crowdsourced / retrieval-based metadata imputation + LALM captioning |

Table 6: Example caption sampled from each sourced dataset.

| Dataset | Example Caption |
|---|---|
| AudioCaps | "Distant traffic sounds followed by a car passing closely." |
| Clotho | "Something is being sanded or dragged, manipulated, scraped." |
| MusicCaps | "This is an advertisement jingle music piece. It is an instrumental piece. The main theme is being played by the piano while there is a synth string sound in the melodic background. There is an emotional, heart-touching atmosphere. This piece could be used in the soundtrack of a drama movie during scenes of tragedy. It could also work well as an advertisement jingle where there is an attempted appeal to emotion." |
| WavCaps | "Music is playing while people are walking and crickets are chirping." |
| AudioSetCaps | "A choir performs a folk music piece, utilizing only their voices as instruments. The harmonious and uplifting sounds create an engaging and captivating listening experience." |
| FusionAudio | "A full choir is singing with powerful harmonized vocals" |
| JamendoMaxCaps | "The music is instrumental with a dominant piano sound, falling under the genres of ambient, classical, and contemporary. It carries a mood that is nostalgic and romantic, played in a 4/4 time signature at a tempo of 81.1 bpm. The piano piece evokes a sense of tranquility, making it suitable for scenarios depicting love scenes or peaceful moments in movies." |
| ParaSpeechCaps | "A male speaker delivers his words quickly with a medium-pitched voice. His speech exhibits a flowing rhythm and is recorded in an environment that is balanced in clarity. There is a subtle nasal quality to his speech, suggesting an American accent." |

## A.2 SOURCED DATASETS FOR CAPTIONSTEW

CaptionStew aggregates eight open-source audio caption datasets to address data scarcity and limited diversity in current audio-language pretraining. The constituent datasets span environmental sounds, music, and expressive speech, with fundamentally different captioning approaches—from crowdsourced human annotation to expert curation to various LLM-based generation pipelines. Table 5 and Table 6 detail each dataset's characteristics and provide example captions that illustrate the diverse descriptive styles, ranging from concise event descriptions to detailed multi-sentence narratives with fine-grained acoustic and contextual information. During aggregation, we filter audio samples longer than one minute for computational efficiency and remove samples that overlap with common audio understanding benchmarks (Kim et al., 2019; Drossos et al., 2020; Kim et al., 2019; Agostinelli et al., 2023; Fonseca et al., 2021; Chen et al., 2020a; Salamon et al., 2014) to prevent data leakage. This approach preserves the unique characteristics of each source while creating a unified corpus that captures broader semantic coverage than individual datasets.

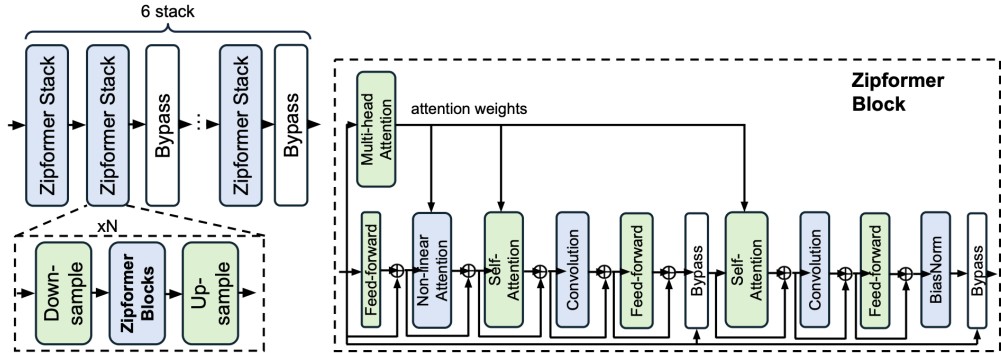

Figure 4: Model diagram of Zipformer.

## A.3 ZIPFORMER MODEL

In this work, we adopt the Zipformer-M architecture (Yao et al., 2024) as the audio encoder, chosen for its memory efficiency on long sequences and strong performance across audio tasks. The architecture employs a U-Net-inspired design with six Transformer stages that process sequences at multiple temporal resolutions. The stages operate at progressively decreasing then increasing frame rates (50, 25, 12.5, 6.25, 12.5, and 25 Hz), with residual and upsampling connections between stages to capture both fine-grained and long-range temporal patterns.

We implement the original 2,2,3,4,3,2 block configuration, where each number indicates the blocks per stage. After processing through all stages, outputs are fused at 25 Hz to produce frame-level embeddings. The model incorporates several architectural improvements from the original work: Bias-Norm for gradient stability over long sequences, Swoosh activation functions for better convergence, and compatibility with the ScaledAdam optimizer. The resulting embeddings are 768-dimensional and used consistently across all downstream evaluation tasks.

Although Zipformer was originally designed for automatic speech recognition, we conducted preliminary experiments to validate its effectiveness as a general audio encoder across diverse domains. As in Table 7, our initial studies confirmed that Zipformer achieves competitive performance on environmental sound classification, music understanding, and speaker-related tasks, demonstrating its suitability as a unified backbone for multi-domain audio representation learning. This cross-domain efficacy makes it an appropriate choice for our audio-language pretraining experiments that span speech, music, and environmental audio.

Table 7: Zipformer performance across audio domains when trained from scratch on individual datasets, demonstrating cross-domain efficacy as a general audio encoder.

| AudioSet (mAP) | VggSound (acc) | VoxCeleb2 (acc) | CREMA (acc) | MagnaTagATune (mAP) | NSynth-Instrument (acc) |
|---|---|---|---|---|---|
| 0.46 | 54.2 | 84.8 | 65.4 | 0.38 | 78.8 |

## A.4 EVALUATION DATASETS

Table 8 details the evaluation datasets and their metrics used for assessing audio representation quality across our three evaluation protocols: linear probing Fonseca et al. (2021); Chen et al. (2020a); Chung et al. (2018); Cao et al. (2014); Law et al. (2010); Engel et al. (2017); Hershey et al. (2021); Ebbers et al. (2022), audio-language alignment Kim et al. (2019); Diwan et al. (2025); Agostinelli et al. (2023); Lin (2004) and open-form question answering Lipping et al. (2022); Liu et al. (2024); Yang et al. (2024b).

Table 8: Details of the dataset used for assessing audio representation. [†]evaluate by GPT-4 in AIR-Bench. [‡]synthesized with public available speech datasets (Ardila et al., 2019; Busso et al., 2008; Cao et al., 2014; Livingstone & Russo, 2018; Poria et al., 2018) with fixed question template.

| Evaluation Dataset | Task | #samples | #class | train | eval | Metrics |
|---|---|---|---|---|---|---|
| FSD-50k | Multi-label audio event classification | 37,168 / 10,231 | 200 | ✓ | ✓ | mAP |
| VggSound | Single-label audio event classification | 183,730 / 15,446 | 309 | ✓ | ✓ | accuracy |
| VoxCeleb2 | Speaker identification | 1,092,009 / 36,693 | 5,994 | ✓ | ✓ | accuracy |
| CREMA-D | Speech emotion recognition | 6,030 / 706 | 6 | ✓ | ✓ | accuracy |
| MagnaTagATune | Music tagging | 19,425 / 4,856 | 50 | ✓ | ✓ | mAP |
| NSynth | Musical instrument classification | 289,205 / 4,096 | 11 | ✓ | ✓ | accuracy |
| AudioSet-strong | Sound event detection | 103,463 / 16,996 | 456 | ✓ | ✓ | PSDS1 |
| AudioCaps | Text-to-audio retrieval | 49,838 / 975 | – | ✓ | ✓ | Recall@1 |
| ParaSpeechCaps | Audio captioning | 116,516 / 500 | – | ✓ | ✓ | RougeL |
| MusicCaps | | 2,663 / 500 | – | ✓ | ✓ | |
| ClothoAQA | Open-formed question answering | 7,044 | – | ✓ | ✗ | |
| In-house SpeechQA[‡] | | 160,000 | – | ✓ | ✗ | |
| MusicQA | | 70,011 | – | ✓ | ✗ | |
| AIRBench-chat-sound | | 400 | – | ✗ | ✓ | Score[†] |
| AIRBench-foundation-emotion | | 1,000 | – | ✗ | ✓ | |
| AIRBench-foundation-gender | | 1,000 | – | ✗ | ✓ | |
| AIRBench-foundation-age | | 1,000 | – | ✗ | ✓ | |
| AIRBench-chat-sound | | 400 | – | ✗ | ✓ | |

## A.5 MAIN RESULTS (CONT.)

Table 9 presents linear probing results when using multi-head attention pooling instead of mean pooling. With learned attention pooling, the performance gap between contrastive and captioning objectives narrows substantially, particularly evident on speaker identification where captioning-scratch achieves 72.86% compared to 46.67% with mean pooling (Table 3). This demonstrates that captioning models benefit significantly from adaptive pooling mechanisms, while contrastive learning's explicit optimization for clip-level representations shows less sensitivity to pooling strategy. These results underscore the critical importance of appropriate downstream module selection when evaluating different pretraining paradigms, as the choice of pooling mechanism can dramatically influence conclusions about objective effectiveness. The improved performance across all methods with attention pooling also suggests that frame-level representations from both objectives contain rich information that can be better exploited through learned aggregation. SOTA results and SSL baseline results in Table 3 and Table 9 are quoted collectively from Niizumi et al. (2025); Turian et al. (2022); Li & Li (2022); Wang et al. (2022); Bharadwaj et al. (2025); Gong et al. (2022b); Lanzendörfer et al. (2025); Bai et al. (2025); Yang et al. (2024b).

Table 9: Linear probing results when using multi-head attention pooling.

| Method | Model Initialization | Audio-language Pretraining | linear probing | | | | | |
|---|---|---|---|---|---|---|---|---|
| | | | AEC FSD50k | AEC VggSound | SID VoxCeleb2 | SER CREMA | MTAG MagnaTagATune | INST NSynth |
| *Our Supervised Baselines* | | | | | | | | |
| Zipformer-AEC Yao et al. (2024) | AS SL | – | 0.656 | 56.23 | 58.76 | 72.52 | 0.405 | 67.19 |
| *Our Audio-language Pretrained Models* | | | | | | | | |
| Contrastive-*scratch* | – | CS10M | 0.640 | 52.81 | **72.86** | **74.50** | 0.406 | 75.00 |
| Captioning-*scratch* | – | CS10M | 0.619 | 50.97 | 56.64 | 70.40 | 0.406 | 72.10 |
| Contrastive-*init* | AS SL | CS10M | **0.670** | 54.89 | 72.24 | 73.09 | **0.412** | **76.70** |
| Captioning-*init* | AS SL | CS10M | 0.660 | 53.68 | 62.24 | 71.67 | 0.411 | 74.49 |
| SOTA[‡] | | | 0.655 | 59.50 | 96.20 | – | 0.414 | 79.20 |

## A.6 ADDITIONAL RESULTS

Aside from learning representations, we also compare against state-of-the-art audio-text retrieval models to assess our approach's performance on the specific task it was designed for. Table 10 presents retrieval results for our best-performing model (Contrastive-init) against state-of-the-art audio-text retrieval model (Bai et al., 2025). Our model achieving comparable or superior results on benchmarks in various audio domains, with particularly strong performance on speech and music retrieval. The results indicate that our general-purpose audio-language pretraining approach can compete with specialized retrieval models while offering broader applicability across diverse usage scenarios.

Table 10: audio-text retrieval of the best performing model (Contrastive-init) against state-of-the-art audio-text retrieval model. [†]reproduce by ourselves.

| Model | Text-to-audio | | | Audio-to-text | | |
|---|---|---|---|---|---|---|
| | AudioCaps | ParaSpeechCaps | MusicCaps | AudioCaps | ParaSpeechCaps | MusicCaps |
| AudioSetCaps[†] | 49.7 / 79.2 | 0.8 / 2.5 | 13.4 / 30.6 | 45.9 / 80.8 | 0.2 / 3.8 | 12.0 / 29.0 |
| Contrastive-init (ours) | 44.4 / 79.0 | 29.6 / 61.6 | 22.4 / 53.0 | 47.2 / 78.8 | 27.0 / 57.4 | 26.0 / 56.2 |

## A.7 THE USE OF LARGE LANGUAGE MODEL

The authors used large language models to assist with writing refinement and grammatical corrections during the drafting process. All technical content, experimental design, analysis, and conclusions remain the authors' original contributions.

