# OpenReview forum: "Revisiting Audio-language Pretraining for Learning General-purpose Audio Representation"
_ICLR.cc/2026/Conference — ICLR 2026 Conference Withdrawn Submission_

### Official Review · Reviewer_JT7u · 2025-11-01

**Soundness:** 4
**Presentation:** 3
**Contribution:** 2
**Rating:** 4
**Confidence:** 4

**Summary:**

This paper presents a comprehensive empirical study revisiting audio-language pretraining as a pathway to general-purpose audio representations. The authors identify key challenges in the field: data scarcity, limited caption diversity, and a narrow focus on contrastive learning for retrieval. To address these, they introduce CaptionStew, a large-scale (10.7M captions, 9.3M audio samples) aggregation of diverse, open-source audio-text datasets spanning environmental sounds, music, and speech. Using this dataset, they conduct a systematic comparison of two pretraining objectives: the dominant contrastive learning approach and an underexplored captioning (generative) objective. The evaluation is extensive, covering linear probing on discriminative tasks, audio-language alignment (retrieval, captioning), and open-form question answering. Key findings include: 1) Contrastive learning is more data-efficient and excels at discriminative tasks, while captioning shows promise for language-involved tasks and better scalability in some settings; 2) Supervised initialization from AudioSet provides benefits but diminishes for tasks unrelated to its ontology; 3) Audio-language pretraining can produce competitive, transferable representations across diverse audio domains, establishing its viability for general-purpose audio representation learning.

**Strengths:**

1. The systematic and comprehensive evaluation is a major strength of the paper. The authors go beyond the standard audio-text retrieval benchmark and evaluate representations across a wide suite of tasks (audio event classification, speaker ID, emotion recognition, music tagging, sound event detection, QA) and protocols (linear probing, retrieval, captioning, LLM-based QA). This provides a holistic and convincing view of the models' capabilities and transferability.
2. The paper provides the first systematic, controlled comparison of contrastive and captioning objectives for audio representation learning. The insights are well-supported by the results across different evaluation protocols and data scales.
3. The commitment to releasing data preparation recipes, training scripts, evaluation protocols, and pretrained models is highly commendable and will significantly accelerate future research in this area.

**Weaknesses:**

1. The work's foundational elements lack significant innovation. The CaptionStew dataset, while large-scale and practically useful, is an aggregation of existing public datasets without a novel data creation or curation methodology. Similarly, the two primary technical insights—that contrastive learning is highly data-efficient and that generative (captioning) objectives scale well with more data—are well-established principles in machine learning, particularly in vision and language domains. Their demonstration in the audio-language context is valuable but does not constitute a novel finding. The observation that the benefits of supervised pre-training diminish with scaling data also aligns with broader trends in representation learning.
2. The paper's primary contribution is a thorough empirical study rather than a methodological breakthrough. It expertly combines existing techniques (contrastive loss, mixed autoregressive/parallel decoding from CapPa) and resources (public datasets) to perform a systematic comparison. While this is highly valuable for the community, it means the paper does not introduce a model, training method, data (aggregation of existing public datasets), or insights (most are common sense).

**Questions:**

1. You selected Zipformer for its efficiency and preliminary cross-domain performance. Did you conduct ablation studies with other popular audio backbones (e.g., HTSAT, BEATs, EAT) to ensure the observed conclusions are not architecture-specific? Would you expect the ranking between contrastive and captioning objectives to change with a different encoder?
2. The paper presents a clear dichotomy between contrastive and captioning objectives. It does not explore hybrid approaches (e.g., multi-task learning combining both objectives). Will it show promise and could it combine the strengths of both paradigms?

---

> ### Author Response · Authors · 2025-11-26
> **Response to Official Review by Reviewer JT7u (1/2)**
>
> **Q0:** Overall concern on novelty
>
> **A0:** We appreciate the reviewer’s acknowledgement on the value of this work and thoughtful assessment of the technical novelty raised in the weaknesses section. Here we respectfully reiterate that our goal is not to present a new architecture, a novel data-generation pipeline, or a fundamentally new learning objective; rather, the work is motivated by **a critical gap that has remained unaddressed** in the audio and speech processing community.
>
> Unlike vision–language research—where contrastive learning [1,5], captioning-based objectives [2,3], and large-scale empirical analyses [4,6] have converged toward well-understood representation learning frameworks—**audio–language pretraining is still in an early developmental stage** in this context. Despite recent progress, audio–language pretraining has been used primarily for retrieval [7,8,9], and the field lacks a systematic understanding of whether it can serve as a general-purpose representation learning framework. In particular, the community does not yet know how different objectives behave across data scales, or how audio–language pretraining transfers across heterogeneous audio domains such as speech, music, and environmental sound. This absence of empirical foundations has led to **inconsistent design choices and uncertainty across the field**.
>
> Crucially, trends from vision or text cannot be assumed to transfer to audio without explicit evidence; doing so overlooks modality-specific differences and risks drawing unsupported conclusions. Audio exhibits fundamentally different charateristics (e.g., temporal continuity, polyphony, and semantic granularity), which create fundamentally different learning dynamics. These differences highlight the need for **controlled, modality-specific empirical evidence** rather than extrapolations from other modalities. To our knowledge, no prior work has provided such systematic evidence.
>
> **Our contribution is therefore an empirical study designed to fill this knowledge gap**. We provide the first unified and controlled comparison of two major audio–language pretraining objectives (contrastive vs. captioning) across multiple data scales and across a broad suite of downstream tasks. This study reveals several insights that were previously undocumented in audio-processing literature and, in some cases, diverge from trends reported in other fields. For example:
>  - We provide the first explicit evidence that audio–language pretraining produces competitive general-purpose audio representations comparable to established supervised and self-supervised baselines.
>  - We identify reverse-scaling effects on temporal tasks such as sound event detection, revealing a previously unreported tension between global semantic supervision and temporal localization.
>  - We highlight that existing audio-caption datasets have limited lexical diversity and show how this limitation meaningfully affects representation learning, an aspect not previously examined.
>
> With respect to the dataset, CaptionStew does not claim to introduce new raw audio content. Its contribution lies in offering a **harmonized, de-duplicated, and leakage-checked aggregation** with unified data preparation recipes, enabling controlled scaling experiments (from 0.4M to 10M audio–text pairs). This role is functionally analogous to LAION’s role in vision–language research: not introducing new content, but providing a standardized empirical basis for reproducible and controlled large-scale studies.
>
> We respectfully submit that providing rigorous empirical evidence is a meaningful and necessary contribution, especially in a modality where pretraining is understudied. We believe **establishing a clear empirical foundation aligns with scientific best practices and is essential for guiding future methodological innovations**. We will revise the introduction and dataset sections to more clearly highlight this motivation and to position the work as a foundational empirical study rather than a methodological one.

---

> ### Author Response · Authors · 2025-11-27
> **Response to Official Review by Reviewer JT7u (2/2)**
>
> **Q1:** You selected Zipformer for its efficiency and preliminary cross-domain performance. Did you conduct ablation studies with other popular audio backbones (e.g., HTSAT, BEATs, EAT) to ensure the observed conclusions are not architecture-specific? Would you expect the ranking between contrastive and captioning objectives to change with a different encoder?
>
> **A1:** We thank the reviewer for raising this important point. We chose Zipformer primarily because it offers a competitive accuracy–efficiency trade-off across speech, music, and environmental tasks in our preliminary screening. Its hybrid convolution–attention design for aggregating temporal information should be broadly suitable for multi-domain audio processing rather than being specialized for one task family. We agree that, ideally, one would replicate all experiments across several encoders; however, conducting full 10M-pair pretraining runs on HTSAT/BEATs/EAT is beyond the computational budget of this work. That said, several pieces of evidence suggest that our conclusions are not tied to Zipformer's architecture. Across multiple downstream projection heads (mean pooling vs. multi-head attention pooling) and heterogeneous task families, the relative ranking between contrastive and captioning objectives remains consistent: contrastive excels at data-efficient, discriminative linear probes; captioning gains strength on language-involved tasks and improves with scale. This potentially suggests that the observed trade-offs are driven more by the form of supervision than by a particular encoder micro-architecture.
>
> We expect that switching to HTSAT/BEATs/EAT would change the absolute performance but would not invert the core findings. To further support this, we are running a small-scale sanity check using a different encoder on a 400k subset and will report these results in the Appendix.
>
> **Q2:** The paper presents a clear dichotomy between contrastive and captioning objectives. It does not explore hybrid approaches (e.g., multi-task learning combining both objectives). Will it show promise and could it combine the strengths of both paradigms?
>
> **A2:** We agree that hybrid objectives are a natural and promising extension. In this work, we intentionally restricted the study to pure contrastive and pure captioning to isolate the effect of each supervision signal under controlled conditions. Again, our goal was to fill a key empirical gap in the literature: there is currently no systematic comparison of these objectives in audio–language pretraining, and hybrid approaches would make interpretation substantially more difficult.
>
> We will make this scope choice more explicit in the paper and add a discussion outlining hybrid designs as a valuable research direction. Prior results in vision–language pretraining (e.g., models that combine contrastive and generative losses [10]) suggest that multi-task objectives can further improve performance. Based on our findings, we believe similar benefits are likely for audio–language models, and we view hybrid losses, along with other potential combination (e.g. self-supervised loss), as a promising next step that can directly build on the empirical foundation established in this work.
>
> **Reference**
> [1] Radford, Alec, et al. "Learning transferable visual models from natural language supervision." International conference on machine learning. PmLR, 2021.
> [2] Zhai, Xiaohua, et al. "Sigmoid loss for language image pre-training." Proceedings of the IEEE/CVF international conference on computer vision. 2023.
> [3] Tschannen, Michael, et al. "Image captioners are scalable vision learners too." Advances in Neural Information Processing Systems 36 (2023): 46830-46855.
> [4] Fini, Enrico, et al. "Multimodal autoregressive pre-training of large vision encoders." Proceedings of the Computer Vision and Pattern Recognition Conference. 2025.
> [5] Liu, Haotian, et al. "Visual instruction tuning." Advances in neural information processing systems 36 (2023): 34892-34916.
> [6] Li, Bo, et al. "LLaVA-OneVision: Easy Visual Task Transfer." Transactions on Machine Learning Research.
> [7] Wu, Yusong, et al. "Large-scale contrastive language-audio pretraining with feature fusion and keyword-to-caption augmentation." ICASSP 2023-2023 IEEE International Conference on Acoustics, Speech and Signal Processing (ICASSP). IEEE, 2023.
> [8] Mei, Xinhao, et al. "Wavcaps: A chatgpt-assisted weakly-labelled audio captioning dataset for audio-language multimodal research." IEEE/ACM Transactions on Audio, Speech, and Language Processing 32 (2024): 3339-3354.
> [9] Bai, Jisheng, et al. "Audiosetcaps: An enriched audio-caption dataset using automated generation pipeline with large audio and language models." IEEE Transactions on Audio, Speech and Language Processing (2025).
> [10] Tschannen, Michael, et al. "Siglip 2: Multilingual vision-language encoders with improved semantic understanding, localization, and dense features." 2025.

---

### Official Review · Reviewer_vsGT · 2025-11-01

**Soundness:** 3
**Presentation:** 3
**Contribution:** 3
**Rating:** 6
**Confidence:** 3

**Summary:**

This work revisits audio-language pretraining as a pathway to general-purpose audio representations. The authors introduce CaptionStew, a large-scale dataset created by aggregating multiple existing audio caption datasets. They conduct a comprehensive empirical study comparing two pretraining objectives—contrastive learning and captioning—across a wide range of downstream tasks. Their experiments systematically evaluate the impact of data scale, model initialization, and objective design. The results demonstrate that audio-language pretraining can produce competitive and transferable representations across speech, music, and environmental sound domains, positioning it as a viable approach for universal audio understanding.

**Strengths:**

The experimental setup is a major strength of this work. The evaluation is extensive, covering linear probing, audio-language tasks, and open-form QA. The controlled experiments on key variables, pretraining objectives (contrastive vs. captioning), data scaling (from 400K to 10M pairs), and initialization (from scratch vs. supervised pretraining)—are exceptionally thorough and provide robust, convincing evidence for the claims. The paper offers several key findings that challenge common practices, such as the diminishing returns of supervised AudioSet initialization on tasks unrelated to its ontology, and the complementary scaling behaviors of contrastive and captioning objectives. These insights are highly valuable for practitioners and researchers in the field.

**Weaknesses:**

The method for creating the core contribution, CaptionStew, is arguably too simplistic. While aggregating existing datasets is a low-cost solution to the data scarcity problem, it inherits the potential quality issues of the source datasets (e.g., LLM-synthesized captions in WavCaps and AudioSetCaps). The authors acknowledge the limited linguistic diversity in the analysis, but the approach feels unrefined and may limit the upper bound of what the pretrained models can learn.

**Questions:**

You identify limited linguistic diversity as a key constraint in CaptionStew. Beyond simple aggregation, what more sophisticated curation strategies (e.g., keyword-based filtering, diversity sampling, or using LLMs to rewrite captions for style variation) did you consider, and why were they not adopted?

Given the varying quality of source datasets (e.g., human-annotated AudioCaps vs. LLM-synthesized WavCaps), did you observe any correlation between the source of a caption and downstream task performance? For instance, do captions from ParaSpeechCaps lead to better performance on speaker-related tasks?

The mixed autoregressive/parallel decoding (inspired by CapPa) is a key component of your captioning objective. Could you ablate the contribution of each mode? Does the parallel decoding, which forces stronger audio dependency, contribute more to the learned representation quality than the standard autoregressive mode?
The results suggest captioning has an advantage in open-form QA, a more generative task. Do you think this is because the captioning objective inherently trains the audio encoder to be more compatible with a language model's reasoning process, or simply that it learns a richer set of semantic concepts?

---

> ### Author Response · Authors · 2025-11-27
> **Response to Official Review by Reviewer vsGT (1/1)**
>
> **Q1:** Overall concern on CaptionStew dataset
>
> **A1:** We appreciate the reviewer’s thoughtful observation. CaptionStew is indeed conceptually simple: it aggregates multiple open-source audio–text corpora into a unified, large-scale resource. To our knowledge, this is **the first effort in the audio processing community** to harmonize these heterogeneous sources into a single, standardized corpus suitable for large-scale empirical studies. The simplicity is intentional. Our central goal in this paper is not to introduce a new data-generation method, but rather to establish a controlled, replicable, and transparent **empirical foundation for comparing pretraining objectives at scale**. More sophisticated curation strategies may indeed be beneficial, yet **it is not guaranteed**. Therefore, we intentionally start from the simplest, least-assumptive baseline to provide a clean foundation for the field. We explicitly position CaptionStew as a baseline resource, not an end-state. In the revised version, we will expand the discussion of alternative curation strategies, clarify why they were not adopted in the current study, and outline how they constitute promising avenues for future work once the fundamental behaviors of different objectives are more fully understood.
>
> **Q2:** Given the varying quality of source datasets (e.g., human-annotated AudioCaps vs. LLM-synthesized WavCaps), did you observe any correlation between the source of a caption and downstream task performance? For instance, do captions from ParaSpeechCaps lead to better performance on speaker-related tasks?
>
> **A2:** Yes, this is one of the motivations for constructing CaptionStew as a heterogeneous aggregation rather than relying on a single-domain corpus. As shown in our dataset section (Section 3, Appendix A.2), different sources exhibit distinct linguistic focuses. These differences create complementary supervision signals and support better cross-domain generalization compared with models pretrained on a single dataset (Table 3, compared to *Zipformer-AEC*).
>
> We also observe indications of source–task correlations in section 4.5. The data-scaling experiments show that tasks with limited linguistic diversity (e.g., emotion recognition, instrument classification) scale less favorably, reinforcing the link between caption characteristics and downstream task behavior.
>
> **Q3:** The mixed autoregressive/parallel decoding (inspired by CapPa) is a key component of your captioning objective. Could you ablate the contribution of each mode? Does the parallel decoding, which forces stronger audio dependency, contribute more to the learned representation quality than the standard autoregressive mode? The results suggest captioning has an advantage in open-form QA, a more generative task. Do you think this is because the captioning objective inherently trains the audio encoder to be more compatible with a language model's reasoning process, or simply that it learns a richer set of semantic concepts?
>
> **A3:** Our statement regarding the mixed decoding mode is based on preliminary experiments in which the mixed autoregressive/parallel decoder produced stronger downstream results than a purely autoregressive decoder. We are currently *re-running* these ablations under stricter controls and will include the quantitative results in the Appendix. We agree that understanding the relative contribution of each decoding mode is important, and we have clarified this point in the revised text (section 2).
>
> Regarding the captioning objective’s advantage in open-form QA: our interpretation is that captioning encourages the encoder to produce representations that are more naturally compatible with language-model–driven reasoning because the model learns to predict structured, token-level semantic content (text). However, we acknowledge that our evidence is suggestive rather than conclusive.

---

### Official Review · Reviewer_Mit7 · 2025-11-03

**Soundness:** 2
**Presentation:** 1
**Contribution:** 1
**Rating:** 2
**Confidence:** 4

**Summary:**

The paper explores the potential of audio-language pretraining (ALP) as a pathway to develop general-purpose audio representations that can transfer effectively across diverse domains, including speech, music, and environmental sounds. The authors argue that ALP—by grounding audio perception in natural language—offers a unified learning framework with strong promise, though existing models have primarily excelled in retrieval tasks and lag behind vision-language models like CLIP in serving as general-purpose encoders. They identify three key limitations hindering progress in ALP: the scarcity of large-scale audio-text corpora, limited caption diversity, and the absence of systematic evaluations of pretraining objectives.

To address these challenges, the authors introduce CaptionStew (CS10M), a large-scale dataset comprising 10.7 million captions paired with 9.3 million audio samples spanning over 37,000 hours across multiple domains. CaptionStew aggregates and harmonizes several open-source corpora such as AudioCaps, WavCaps, MusicCaps, and ParaSpeechCaps, thereby enhancing the diversity of captioning styles—from concise human annotations to detailed, LLM-generated narratives and domain-specific descriptors. Although CaptionStew significantly broadens the vocabulary (56,586 unique words), its lexical diversity remains lower than that of mature image-text corpora, potentially constraining scaling gains for certain fine-grained tasks.

The paper also presents a systematic comparison of pretraining objectives, contrasting contrastive learning (a discriminative, InfoNCE-based approach) with captioning objectives (generative, via autoregressive or parallel prediction). Contrastive learning demonstrates superior data efficiency at smaller scales and stronger performance on linear probing tasks such as audio event classification and speaker identification. In contrast, the captioning objective scales better with larger datasets and performs slightly better on language-involved tasks, such as open-ended question answering and caption generation. Notably, the performance gap between these objectives narrows considerably when adaptive pooling mechanisms (e.g., multi-head attention pooling) replace mean pooling, underscoring the influence of downstream architectural choices.

Further experiments examine the effects of initialization and data scaling. While initializing the audio encoder with supervised models pretrained on AudioSet (AS SL) yields immediate performance boosts—particularly for discriminative objectives—these benefits diminish or vanish at larger data scales. This suggests that diverse caption-based supervision can rival or even surpass domain-specific supervised initialization when training for general-purpose representation learning. Scaling analyses reveal steady performance improvements across most tasks with increasing data, except in sound event detection, where performance declines as caption data grows, possibly due to conflicts between global semantic supervision and the temporal precision required for event localization.

In conclusion, the study demonstrates that audio-language pretraining produces competitive and transferable audio representations, marking it as a promising strategy toward universal audio understanding. Conceptually, general-purpose audio representation can be likened to a universal dictionary for sounds: while traditional supervised methods create specialized, domain-limited dictionaries, ALP—empowered by diverse descriptive language from CaptionStew—builds a flexible, semantic scaffold capable of defining both a "voice timbre" and a "rhythmic pattern." In this analogy, contrastive learning efficiently constructs a strong core vocabulary, whereas generative captioning gradually cultivates a broader and more scalable comprehension of sound when given sufficient data.

**Strengths:**

The paper demonstrates several notable strengths, particularly through its systematic approach to addressing existing limitations and its comprehensive empirical validation of audio-language pretraining (ALP) as a pathway to developing general-purpose audio representations. One of its most significant contributions is the introduction of the CaptionStew (CS10M) dataset, a large-scale and diverse resource specifically designed to overcome the scarcity of extensive audio-text corpora that has long hindered progress in ALP. CaptionStew comprises 10.7 million captions paired with 9.3 million audio samples, totaling over 37,000 hours of audio—an order of magnitude larger than prior datasets. Its diversity stems from the aggregation of multiple open-source corpora across speech, music, and environmental sounds, coupled with varying captioning styles, including human-annotated descriptions, LLM-generated narratives, and fine-grained annotations of speaking or musical attributes. The paper further validates this diversity through t-SNE visualizations of sentence embeddings, demonstrating the unique descriptive focuses of each corpus and highlighting how their complementary linguistic perspectives enrich model generalization.

Beyond data creation, the study conducts the first systematic comparison between contrastive and captioning pretraining objectives across multiple audio domains. This comprehensive evaluation reveals clear complementary strengths: contrastive learning proves more data-efficient and excels in linear probing tasks like classification, whereas captioning objectives exhibit better scalability and stronger performance in language-dependent tasks such as open-form question answering. Importantly, the authors show that the performance gap between these objectives narrows considerably when adaptive pooling mechanisms, such as multi-head attention pooling, are used instead of simpler mean pooling—underscoring the critical role of downstream architecture in fair representation assessment.

The paper also challenges common practices in model initialization, particularly the reliance on supervised pretraining from datasets such as AudioSet (AS SL). While supervised initialization offers early-stage benefits, the results demonstrate that these advantages diminish or vanish entirely at larger data scales. Moreover, such pretraining can bias models toward event-level semantics, potentially hindering performance on tasks requiring fine-grained acoustic understanding, such as speaker identification or music tagging. This finding questions the necessity of domain-specific supervision and highlights the potential of large-scale, diverse caption data to serve as a self-sufficient foundation for semantic learning.

**Weaknesses:**

A major contribution of the paper, the CaptionStew (CS10M) dataset, also represents a few limitation. While its scale and diversity mark a substantial advance for audio-language pretraining, the authors recognize several issues inherent in its design. First, CaptionStew’s reliance on LLM-synthesized and augmented captions introduces potential biases and artifacts, as no extensive human verification or quality control was conducted across the dataset. This could affect the fidelity and reliability of learned representations. Second, despite an expanded vocabulary of over 56,000 unique words, the dataset exhibits low lexical diversity overall. Aggregating multiple corpora does not inherently ensure rich linguistic variation, and CaptionStew’s Distinct-n metrics remain considerably lower than those observed in mature image-text datasets. This limitation, partly attributable to certain constituent datasets with low linguistic diversity, may have contributed to weaker scaling behavior in tasks like emotion recognition and musical instrument classification, where nuanced language is essential for robust representation learning.

Beyond dataset concerns, the paper’s technical novelty is relatively limited. The work primarily serves as an empirical and systematic study, combining existing methods such as contrastive learning, captioning objectives, and large-scale data aggregation rather than introducing fundamentally new algorithms or architectures. Techniques like the mixed autoregressive/parallel training approach for captioning are adapted directly from prior vision-language models (e.g., CapPa), and the architectural foundations follow well-established conventions. The authors justify this approach by emphasizing their aim to provide a comprehensive comparative analysis rather than a methodological innovation—an important contribution in a field that previously lacked rigorous cross-objective evaluations.

Finally, the study’s scalability constraints limit the generalizability of its conclusions. The experiments are conducted on a dataset of roughly 10 million audio-text pairs using relatively modest model sizes, far smaller than the billion-scale datasets and architectures typical in modern vision-language research. As a result, the findings may underestimate the full potential of audio-language pretraining, particularly for the captioning objective, which appears to benefit disproportionately from increased scale. The authors note that achieving parity between captioning and contrastive methods on certain benchmarks may require scaling to hundreds of millions of audio-text pairs. Additionally, the study does not explore more advanced architectures or integration with large language models, which could further enhance performance. These omissions stem not from oversight but from deliberate experimental control and computational limitations.

**Questions:**

Thanks for the paper, let's work on this paper collectively to make it a better publication. Kindly address all of these questions.

1. [Line 50] : "diverse audio modalities" - A bit vague, define what you wish to mention, what kind of diversity are you refereeing? And what are different audio modalitties?

2. [Line 60] : audio-language pertaining : I recommend you to cite few other papers which are there in this field (Have previously reviewed these work) : 1) Sinha, Anshuman, et al. "Enhancing Audio-Language Models through Self-Supervised Post-Training with Text-Audio Pairs." arXiv preprint arXiv:2408.09269 (2024)., 2) Ghosh, Sreyan, et al. "Compa: Addressing the gap in compositional reasoning in audio-language models." arXiv preprint arXiv:2310.08753 (2023).

3. [Line 72] : Define the audio spectrum which you wish to mention.

4. [Line 83-96] : Refers the sections, figures, plots where you have addressed these four points in each bullet point. It helps the readers.

5. [ Line 141] : For captioning objective, how are you stating that this is a general objective? Do you think auto-regressive generation of captions from audio is a more general way of learning representations than a contrastive model? How do you define "generality" in terms of learning?

Tbh, the auto-regressive training which you're implementing is actually an imitation learning method (supervised learning). Do you think this method is more general than a self-supervised contrastive learning method?

6. [Line 143] : denser token-level supervision? What do you mean by density in this context? How do you claim this?

7. [Line 160] : Parallel mode enforces stringer dependency ... ; Is this a claim or a proposition? If it's a claim, do you have results supporting this claim? If it's your proposition, then what makes you can write "we assume/ think .... ; and further provide evidence of this effect; although the evidence need not be strong".

8. [Line 180-205] : Can you please get a figure of this section and replace the long texts; Usually other good publications have demonstrated a figure for their data strategy. Makes the life easier for a reader.

9. [Line 185] : What do you mean by single generation pipeline?

10. [Line 203] : Since this pre-training is done over a huge dataset which covers different aspects of audio, how do you know whether it's your model strategy or just the data coverage?

11. [Table 3] :

(a) What is AEC, SID, etc? The core results of the model look a bit on the weaker side; You've compared the strengths of your work with a baseline (Zipformer) on which you've further post-trained on specialized datasets; right? Did we expect any other result than the one which have been posted?

(b) How do these results support your initial claim of making a "genera" purpose audio encoder (which essentially learn a general purpose audio representation.)?

(c) Put Section 4.4 in bullets; the crux of the paper shouldn't be in long paragraphs.

12. Where is the ablation study which compares the effects of adding different data sources, as you mentioned something about adding different kinds of data; now what effects does it have on your model? Or did you just simply added a mix and expected everything to work? (If we understand deep how deep learning works, a mix of everything would more often than not work; but your paper is not aimed just at that. You wish to show how to make a general purpose audio encoder.)

13 [Figure 3] The t-SNE plot show some grouping, but it does not reflect anything on your learning! Moreover do you have any data on how separate the groups are? Because to me only two of the group (JamendoMaxCaps and ParaSpeechCaps seem distinct).

That's all for now, let's discuss the above. All the best

---

> ### Author Response · Authors · 2025-11-26
> **Response to Official Review by Reviewer Mit7 (1/4)**
>
> **Weakness 1:** The limitation of CaptionStew dataset
>
> **Response 1:** We appreciate the reviewer highlighting CaptionStew’s scale and diversity. We would like to clarify two points. First, the statement that the dataset lacks human verification or quality control is inaccurate. CaptionStew aggregates existing public datasets whose captions have already undergone **careful curation or extensive evaluation in their original work** (e.g., AudioCaps, Clotho, MusicCaps, WavCaps, AudioSetCaps, etc.). Our contribution lies in harmonizing, de-duplicating, leakage-checking, and unifying these sources to support controlled empirical comparisons, rather than generating new synthetic captions.
>
> Second, we did not claim that “biases and artifacts” from each individual datasets degrades the reliability of the learned representations. As stated in Section 3, these issues refer primarily to the narrow descriptive focus and limited linguistic style diversity within each single caption-generation pipeline. This limited diversity constrains the semantic richness available during training, which in turn restricts the potential of captions to serve as a flexible, domain-agnostic semantic manifold. Addressing this limitation is precisely the motivation for creating CaptionStew, which brings together complementary linguistic styles from multiple sources to broaden coverage and reduce single-source bias.
>
> **Weakness 2:** Limitation on technical novelty
>
> **Response 2:** We appreciate that the reviewer recognizes this work as “an important contribution in a field that previously lacked rigorous cross-objective evaluations.” We reiterate that our primary aim is not to introduce a new architecture, but to address **the fundamental absence of systematic empirical evidence** in audio–language pretraining. Prior works in audio and speech do not examine or quantify:
>  - whether audio–language pretraining can serve as a viable pathway toward general-purpose audio representation learning,
>  - how different objectives (contrastive vs. captioning) shape the learned representations
>  - cross-domain behavior across speech, music, and environmental sounds on diverse task families using a single audio encoder
>  - scaling behavior from 0.4M → 10M pairs across all tasks
>  - when supervised initialization from AudioSet helps or, unexpectedly, hurts
>  - and how linguistic diversity (or lack thereof) affects downstream performance.
> These are new, non-trivial findings specific to the audio modality and cannot be inferred from other fields without concrete evidence. Large-scale empirical studies are valued contributions when they shift a community’s understanding of design choices, and our study is designed precisely to fill this gap for audio and speech research.
>
> **Weakness 3:** Limited model size and dataset scale may underestimate the results
>
> **Response 3:** We agree that exploring larger models and even broader datasets is an important direction for future work. At the same time, CaptionStew is already, to the best of our knowledge, the largest audio–caption corpus currently available. Within this constraint, our study provides the first scaling analysis across a wide variety of tasks spanning speech, music, and environmental sounds. Importantly, our goal is not to maximize raw performance but to establish controlled, causal insights into objective design, scaling trends, and cross-domain generalization. These are foundational questions that must be answered before scaling to 100M+ pairs or billion-parameter models.
> While compute constraints remain a practical limitation in audio research, they do not diminish the contribution of providing a rigorously characterized empirical foundation for future audio–language pretraining. Our findings directly inform how larger-scale audio-language pretraining systems should be designed and evaluated.

---

> ### Author Response · Authors · 2025-11-26
> **Response to Official Review by Reviewer Mit7 (2/4)**
>
> **Q1:** [Line 50] "diverse audio modalities" - A bit vague, define what you wish to mention, what kind of diversity are you refereeing? And what are different audio modalitties?
>
> **A1:** Thank you for pointing this out. This phrase was a typo. Our intended wording was: “chieving general-purpose audio representations that transfer robustly across **diverse audio processing tasks** remains a challenging and actively pursued goal in the field.” We have revised the text accordingly.
>
> **Q2:** [Line 60] I recommend you to cite few other papers which are there in this field (Have previously reviewed these work)
>
> **A2:** Thank you for the suggestions. Although these works primarily focus on compositional and temporal aspects of audio captioning, which differs from our focus, they nevertheless relate to audio–language modeling. We have added them to the related works section to broaden the contextual coverage.
>
> **Q3:** [Line 72] Define the audio spectrum which you wish to mention
>
> **A3:** We appreciate the request for clarification. In the original text, “audio spectrum” was intended to describe the range of semantic attributes that audio signals may carry. To avoid confusion with frequency-domain terminology, we have replaced “spectrum” with “range.”
>
> **Q5:** [ Line 141] : For captioning objective, how are you stating that this is a general objective? Do you think auto-regressive generation of captions from audio is a more general way of learning representations than a contrastive model? How do you define "generality" in terms of learning? Tbh, the auto-regressive training which you're implementing is actually an imitation learning method (supervised learning). Do you think this method is more general than a self-supervised contrastive learning method?
>
> **A5:** We would like to clarify that categorizing captioning as “supervised learning” while treating contrastive learning as self-supervised **is inaccurate in this context**. Both objectives rely on externally provided captions (human-annotated or LLM-generated), and thus both fall under weakly supervised learning using audio–text pairs. Importantly, we do not claim that captioning is inherently “more general” than contrastive learning. Rather, our goal is to compare their different, as they may reflects the complementary learning scheme (discriminative v.s. generative) for aligning audio and language. Finally, our use of “generality” is grounded in empirical evidence: the pretrained encoder is evaluated on **16 diverse downstream tasks** spanning speech, music, environmental sound, and audio–language understanding, and we use this breadth as an operational definition of general-purpose learning, which is a also common practice in prior domain-specific benchmarking efforts [1,2,3].
>
> **Q6:** [Line 143] : denser token-level supervision? What do you mean by density in this context? How do you claim this?
>
> **A6:** By “denser token-level supervision,” we refer to the fact that: First, the cross-attention mechanism in captioning models provides frame-level, temporally fine-grained supervision to the audio encoder. Each generated token attends to the audio sequence, supplying a rich set of localized training signals, in contrast to the utterance-level alignment used in contrastive learning. Moreover, captioning models the joint distribution over all caption tokens, making the learning process more sensitive to fine-grained attributes, relational cues, and word order. In comparison, contrastive learning relies only on global audio–text embeddings and thus behaves similarly to a bag-of-words semantic matcher [4,5]. We have revised the wording in the paper to better convey this intuition.
>
> **Q7:** [Line 160] : Parallel mode enforces stringer dependency ... ; Is this a claim or a proposition?
>
> **A7:** Thank you for asking for clarification. This statement is based on preliminary experiments where the mixed-mode (autoregressive + parallel) captioning objective produced stronger downstream performance than using a purely autoregressive decoder. We are actively rerunning these experiments to report the quantitative results in the final version.
>
> **Q8** [Line 180-205] : Can you please get a figure of this section and replace the long texts; Usually other good publications have demonstrated a figure for their data strategy. Makes the life easier for a reader.
>
> **A8**: We appreciate the reviewer’s suggestion to visualize the data construction process. While our current textual description is already concise and explicit, we will include such a figure in the Appendix sections.

---

> ### Author Response · Authors · 2025-11-26
> **Response to Official Review by Reviewer Mit7 (3/4)**
>
> **Q9:** [Line 185] : What do you mean by single generation pipeline?
>
> **A9:** By “single generation pipeline,” we refer to the fact that captions within each dataset are produced using a consistent annotation procedure (see Appendix A.2)—either (i) human annotation following shared guidelines, or (ii) LLM-based synthesis using a fixed prompting scheme. As a result, captions from the same dataset share a homogeneous linguistic style and descriptive focus. We have revised the text to explain this more clearly and avoid ambiguity.
>
> **Q10:** [Line 203] : Since this pre-training is done over a huge dataset which covers different aspects of audio, how do you know whether it's your model strategy or just the data coverage?
>
> **A10:** We appreciate the reviewer’s question and would like to request clarification, as the intention behind the comment is not fully clear to us. Our work aims to study pretraining objective behavior and scaling effects under a fixed aggregated dataset; the dataset composition itself is held constant across experiments. If the reviewer could elaborate on the specific concern, we would be happy to address it directly.
>
> **Q11:** [Table 3] :
> (a) What is AEC, SID, etc? The core results of the model look a bit on the weaker side; You've compared the strengths of your work with a baseline (Zipformer) on which you've further post-trained on specialized datasets; right? Did we expect any other result than the one which have been posted?
> (b) How do these results support your initial claim of making a "genera" purpose audio encoder (which essentially learn a general purpose audio representation.)?
> (c) Put Section 4.4 in bullets; the crux of the paper shouldn't be in long paragraphs.
>
> **A11:**
> (a) These are standard abbreviations for common audio processing tasks. We now explicitly define all acronyms in Section 4.2. Regarding the baseline: it is not trained on any specialized downstream dataset. The baseline uses AudioSet-supervised pretraining, which is the dominant off-the-shelf encoder in prior work [6,7]. Our comparisons are therefore aligned with established practice. If the reviewer expected different behavior from the baseline, we would appreciate clarification so we may address the concern more precisely.
> (b) As mentioned above, tOur operational definition of “general-purpose” follows established practice in multimodal research: systematic transferability across heterogeneous downstream tasks. We evaluate the encoder on 16 tasks spanning speech, music, environmental sound, event detection, retrieval, captioning, and QA, demonstrating broad transfer capabilities under a unified pretraining setup.
> (c) We appreciate the suggestion. At present, we find that the paragraph structure better preserves the logical flow of our scaling analysis and the interplay between objectives and initialization. Nevertheless, we will consider reformatting if space permits.
>
> **Q12:** Where is the ablation study which compares the effects of adding different data sources, as you mentioned something about adding different kinds of data; now what effects does it have on your model? Or did you just simply added a mix and expected everything to work?
>
> **A12:** Thank you for raising this question. Our work focuses on objective- and scale-level comparisons under a fixed aggregated dataset, rather than on per-source ablations. However, we do include per-corpus lexical diversity analyses and domain-specific evaluations (e.g., music tagging, speaker identification, environmental sound classification), which indirectly reflect the influence of different data sources. Performing full per-source ablations at 10M scale is computationally prohibitive. Nevertheless, we recognize the value of understanding dataset composition effects and are running small-scale per-source ablations to highlight relative contributions. These experiments may not finish before the rebuttal deadline, but we will include findings in the final version if feasible.
>
> **Q13:** [Figure 3] The t-SNE plot show some grouping, but it does not reflect anything on your learning! Moreover do you have any data on how separate the groups are? Because to me only two of the group (JamendoMaxCaps and ParaSpeechCaps seem distinct).
>
> **A13:** We respectfully clarify that Figure 3 is not intended to reflect model learning behavior. Rather, it visualizes pre-training caption embeddings to illustrate the linguistic diversity across source corpora, highlighting stylistic and descriptive differences prior to training. This motivates why aggregating complementary caption sources strengthens the semantic supervision signal. Regarding the remark that “only two groups seem distinct,” this is consistent with our findings: some corpora (e.g., LLM-generated vs. human-annotated) differ more strongly in style, and the t-SNE reflects this accordingly.

---

> ### Author Response · Authors · 2025-11-26
> **Response to Official Review by Reviewer Mit7 (4/4)**
>
> Overall, several of the reviewer’s questions appear to arise from misunderstandings or from interpreting certain figures outside their intended purpose. We have addressed each point above and revised the manuscript to further improve clarity and readability.
>
> **Reference**
> [1] Yang, Shu-wen, et al. "SUPERB: Speech Processing Universal PERformance Benchmark." Proc. Interspeech 2021. 2021.
> [2] Turian, Joseph, et al. "Hear: Holistic evaluation of audio representations." NeurIPS 2021 Competitions and Demonstrations Track. PMLR, 2022.
> [3] Yuan, Ruibin, et al. "Marble: Music audio representation benchmark for universal evaluation." Advances in Neural Information Processing Systems 36 (2023): 39626-39647.
> [4] Yuksekgonul, M., et al. "WHEN AND WHY VISION-LANGUAGE MODELS BEHAVE LIKE BAGS-OF-WORDS, AND WHAT TO DO ABOUT IT?." 11th International Conference on Learning Representations, ICLR 2023. International Conference on Learning Representations, ICLR, 2023.
> [5] Hsieh, Cheng-Yu, et al. "Sugarcrepe: Fixing hackable benchmarks for vision-language compositionality." Advances in neural information processing systems 36 (2023): 31096-31116.
> [6] Gong, Yuan, et al. "Listen, Think, and Understand." International Conference on Learning Representations. 2024.
> [7] Ghosh, Sreyan, et al. "GAMA: A Large Audio-Language Model with Advanced Audio Understanding and Complex Reasoning Abilities." Proceedings of the 2024 Conference on Empirical Methods in Natural Language Processing. 2024.

---

### Official Review · Reviewer_WAVK · 2025-11-04

**Soundness:** 3
**Presentation:** 2
**Contribution:** 2
**Rating:** 2
**Confidence:** 4

**Summary:**

This paper conducts experiments to show that large-scale audio-language pretraining (with contrastive/captioning supervision) can yield good general-purpose audio representations.

However, the technical novelty of this paper is limited. The effort on the dataset is just an aggregation of several existing open-source datasets, and it’s misleading to create a new name for this dataset, as there is no new content being added. The method used in this paper is also of limited novelty. The work in this paper is mostly experimental.

**Strengths:**

1. It's useful to see how model performance scale with the dataset size.

**Weaknesses:**

1. The paper has a lot of missing details in writing:
What do the settings “PSC” and “MC” mean in Table 3.b?
What do “AEC,” “SID,” “SER,” “MTAG,” “INST” and “SED” mean in Table 3.a? Please do not assume readers are already familiar with these acronyms.
2. The paper's experimental setting is unclear. It shows a lot of settings, but the exact setting of the method is not mentioned. Please do not assume reader can understand these setting without explaination.
For example, the author list results of “Contrastive-scratch,” “Captioning-scratch,” etc. in the table, but there are no corresponding explanations of these settings. This can cause confusion for the reader.
3. Please unify the formatting for w2v in Table 3.a
4. The paper mentioned they release the dataset and related model but it’s not found anywhere in the paper.
5. The overall technical contribution is not strong enough.

**Questions:**

Using captioning to perform audio language pretraining is not a common practice. I'm wondering why authors are interested in this scheme, especially considering that captioning pretraining seems to always perform worse than contrastive learning (Figure 2).

---

> ### Author Response · Authors · 2025-11-25
> **Response to Official Review by Reviewer WAVK (1/3)**
>
> **Q0: Overall concern on novelty and contribution**
>
> **A0:** We appreciate the reviewer’s perspective on technical novelty and would like to clarify the motivation and contributions of this work from the standpoint of the audio and speech processing community. Unlike vision–language research—where contrastive [1,5] and captioning pretraining [2,3], along with large-scale, related empirical studies [4,6], have matured into a robust visual representation learning framework—audio–language pretraining remains comparatively underexplored. Crucially, there is still **no consensus on whether audio–language pretraining can serve as a general-purpose representation learning framework**, and existing usage remains largely restricted to retrieval [7,8,9]. The community also lacks a systematic understanding of how **different pretraining objectives behave**, **whether scaling helps uniformly**, and how audio-language pretraining behaves across **heterogeneous audio domains** (speech, music, environmental sounds) and task families. This gap is substantive and real, hindering the progress in the field.
>
> While some may believe that trends from vision or text modalities can be extrapolated to audio, this assumption does not hold. **Audio is not a simple analogue of vision or language**: its temporal structure, event sparsity, polyphony, and semantic granularity create fundamentally different learning dynamics. Therefore, controlled, modality-specific empirical evidence is required, and insights from other fields cannot simply be automatically applied without verification. Establishing such evidence is essential for building a solid foundation for future audio-language pretraining and audio representation learning research.
>
> The goal of our work is therefore not to propose a new architecture or algorithm but to fill the knowledge gap by providing **the first principled, large-scale empirical study** of audio-language pretraining for general-purpose audio representation learning.
> We systematically compare pretraining objectives (contrastive vs. captioning) across multiple data scales and evaluate a single encoder over an unusually broad and representative suite of downstream tasks.
> To our knowledge, no prior work has evaluated one audio encoder simultaneously on speech (speaker identification, emotion recognition), environmental sound classification/detection, music tagging, retrieval, captioning, and open-form question answering under a unified and controlled setup. This comprehensive evaluation yields several actionable insights that were not previously documented in audio literature (just to name a few):
>  - Demonstrating audio-langauge pretraining actually produce competitive audio representation compared to mainstream supervised and self-supervised methods
>  - A strong data-efficiency gap favoring **contrastive learning at small scale**, contrasted with the **scalability and language-alignment advantages of captioning**, which is **an objective that has been largely overlooked**  in audio-language pretraining.
>  - Diminishing (and sometimes negative) effects of AudioSet supervised initialization, contradicting the common practice and long-standing assumption that it is universally beneficial for downstream audio tasks.
>  - Reverse-scaling behavior in temporal tasks such as sound event detection, revealing a fundamental tension between global semantic supervision and temporal localization.
>  - Highlighting the lexical diversity concerns in current audio caption dataset and how it can affect the learned representation, which is never discussed in prior works.
>
> Regarding the dataset: CaptionStew does not claim novel raw content. Its contribution lies in providing a harmonized, de-duplicated, leakage-checked aggregation with unified data recipes, enabling controlled comparisons across scales (0.4M → 10M pairs). It is intended not as a new dataset “brand,” but as a standardized empirical substrate for reproducible ALP research, akin to the role LAION plays in vision–language evaluation.
>
> We respectfully submit that such systematic, modality-grounded empirical work is not only necessary for advancing audio–language modeling but also embodies the foundational scientific principle of **grounding claims or assumptions in empirical evidence**. The absence of clear evidence in this area has hindered understanding, introduced uncertainty in design choices, and slowed research. Our study offers the first rigorously characterized baseline for audio-language pretraining and provides actionable insights that we believe will accelerate future development in building general-purpose audio encoder. We appreciate the reviewer’s thoughtful comments and will further emphasize the importance and contributions of this work in the revised introduction and dataset sections.

---

> ### Author Response · Authors · 2025-11-26
> **Response to Official Review by Reviewer WAVK (2/3)**
>
> **Q1:** The paper has a lot of missing details in writing: What do the settings “PSC” and “MC” mean in Table 3.b? What do “AEC,” “SID,” “SER,” “MTAG,” “INST” and “SED” mean in Table 3.a? Please do not assume readers are already familiar with these acronyms.
>
> **A1:** We have revised Section 4.2 to explicitly define all acronyms used in Tables 3.a and 3.b, avoiding ambiguity. For clarity:
>  - Linear probing tasks: audio event classification (AEC), sound event detection (SED), speaker identification (SID), speech emotion recognition (SER), music tagging (MTAG), and musical instrument classification (INST).
>  - Audio–language alignment evaluation datasets: AudioCaps (AC), ParaSpeechCaps (PSC), and MusicCaps (MC).
>
>
> **Q2:** The paper's experimental setting is unclear. It shows a lot of settings, but the exact setting of the method is not mentioned. Please do not assume reader can understand these setting without explaination. For example, the author list results of “Contrastive-scratch,” “Captioning-scratch,” etc. in the table, but there are no corresponding explanations of these settings. This can cause confusion for the reader.
>
> **A2:** We have revised Section 4.1 to clearly describe all pretraining configurations, specifically defining "*-scratch*" (training from scratch) and "*-init*" (initialized from pretrained checkpoint). The implementation details and distinctions among these settings are now stated explicitly. If additional clarification is needed, we would be happy to further refine this section.
>
> **Q3:** Please unify the formatting for w2v in Table 3.a
>
> **A3:** We have standardized all occurrences of Wav2Vec 2.0 across Table 3.a, Section 4.3, and Section 4.4 to ensure consistent formatting throughout the paper.
>
> **Q4:** The paper mentioned they release the dataset and related model but it’s not found anywhere in the paper.
>
> **A4:** Due to anonymization policy during the review process, we are unable to publicly host the full dataset aggregation or model checkpoints. We have updated the manuscript to clarify this point and will release all artifacts immediately upon acceptance.
>
> **Q5:** The overall technical contribution is not strong enough.
>
> **A5:** Please refer to our detailed response in **A0**, where we clarify the intended contribution of this work and explain its importance for advancing audio–language pretraining.
>
> **Q6:** Using captioning to perform audio language pretraining is not a common practice. I'm wondering why authors are interested in this scheme, especially considering that captioning pretraining seems to always perform worse than contrastive learning (Figure 2).
>
> **A6:** We thank the reviewer for raising this point. While captioning-based pretraining is less common in audio–language research, this is precisely why it merits investigation, particularly given the limited prior work on audio–language pretraining as a whole. Captioning objectives have been extensively studied in vision–language research (e.g., CapPa [2], AIM-v2 [3]) and have shown strong scalability and performance at large data regimes.
>
> We respectfully clarify that captioning does not “always” perform worse: at the current 10M–scale regime, contrastive learning is more data-efficient, but **captioning demonstrates better scalability and stronger performance on language-involving tasks**, as highlighted in Section 1, Section 4.4, and Section 6. These characteristics align with the emerging trend of building large audio–language models.
>
> Taken together, we believe captioning is an important and underexplored objective for advancing audio–language pretraining, and our study provides the first systematic evaluation of its strengths, weaknesses, and scaling behaviors.
>
> [1] Tschannen, Michael, et al. "Image captioners are scalable vision learners too." Advances in Neural Information Processing Systems 36 (2023): 46830-46855.
> [2] Fini, Enrico, et al. "Multimodal autoregressive pre-training of large vision encoders." Proceedings of the Computer Vision and Pattern Recognition Conference. 2025.

---

> ### Author Response · Authors · 2025-11-26
> **Response to Official Review by Reviewer WAVK (3/3)**
>
> **Reference**
> [1] Radford, Alec, et al. "Learning transferable visual models from natural language supervision." International conference on machine learning. PmLR, 2021.
> [2] Tschannen, Michael, et al. "Image captioners are scalable vision learners too." Advances in Neural Information Processing Systems 36 (2023): 46830-46855.
> [3] Fini, Enrico, et al. "Multimodal autoregressive pre-training of large vision encoders." Proceedings of the Computer Vision and Pattern Recognition Conference. 2025.
> [4] Liu, Haotian, et al. "Visual instruction tuning." Advances in neural information processing systems 36 (2023): 34892-34916.
> [5] Zhai, Xiaohua, et al. "Sigmoid loss for language image pre-training." Proceedings of the IEEE/CVF international conference on computer vision. 2023.
> [6] Li, Bo, et al. "LLaVA-OneVision: Easy Visual Task Transfer." Transactions on Machine Learning Research.
> [7] Wu, Yusong, et al. "Large-scale contrastive language-audio pretraining with feature fusion and keyword-to-caption augmentation." ICASSP 2023-2023 IEEE International Conference on Acoustics, Speech and Signal Processing (ICASSP). IEEE, 2023.
> [8] Mei, Xinhao, et al. "Wavcaps: A chatgpt-assisted weakly-labelled audio captioning dataset for audio-language multimodal research." IEEE/ACM Transactions on Audio, Speech, and Language Processing 32 (2024): 3339-3354.
> [9] Bai, Jisheng, et al. "Audiosetcaps: An enriched audio-caption dataset using automated generation pipeline with large audio and language models." IEEE Transactions on Audio, Speech and Language Processing (2025).

---

### Author Response · Authors · 2025-12-03
**Summary of the Work and Rebuttal for the Area Chair**

Dear Area Chair,

We appreciate your time and effort in reviewing our paper. Below we provide a concise summary of our work and the key clarifications made during the rebuttal, in order to assist in your final decision.

**Summary of this work**: We present **the first principled empirical study** designed to address the knowledge gap regarding audio–language pretraining as a general-purpose framework. We introduce CaptionStew, a harmonized aggregation of 10.7M audio–text pairs, serving as a standardized substrate for reproducible research. Using this controlled setup, we conduct a systematic comparison of contrastive versus captioning objectives across 16 downstream audio processing tasks. Our study provides actionable insights by: identifying critical trade-offs between data efficiency and scalability for different pretraining objectives; revealing the diminishing returns of supervised initialization at scale; surfacing the limited lexical diversity in current caption datasets that hinder model performance. We hope the empirical evidence found in the work will accelerate the development of future general-purpose audio representation learning.

**Recognition from reviewers**: While there were initial concerns regarding technical novelty, reviewers recognized the significant value of our rigorous empirical approach and the utility of our resources:

 - Systematic Evaluation: “exceptionally thorough and provide robust, convincing evidence” (Reviewer vsGT), “holistic and convincing view” (Reviewer JT7u), “comprehensive empirical validation” (Reviewer Mit7).

 - Actionable Insights: “key findings that challenge common practices… highly valuable for practitioners” (Reviewer vsGT), “useful to see how model performance scale” (Reviewer WAVK), “insights are well-supported” (Reviewer JT7u).

 - Community Contribution: “significant contributions is the introduction of the CaptionStew” (Reviewer Mit7), “highly commendable” commitment to releasing recipes and models (Reviewer JT7u).

**Key Rebuttal Clarifications**: The primary concern raised was whether the work possessed sufficient "technical novelty" (i.e., new architectures). In our rebuttal, we addressed this by:

 - Reframing the Contribution: We clarified that our goal is not to invent a new method, but to fill a critical knowledge gap for the audio processing community. We argued that providing the first controlled, modality-specific evidence for audio modality is a foundational scientific contribution (addressing WAVK, JT7u, Mit7).

 - Clarifying Dataset Purpose: We emphasized that CaptionStew is designed for reproducibility and standardization (similar to LAION’s role in vision), rather than just raw data aggregation (addressing Mit7, vsGT).

 - Technical Details: We resolved specific questions regarding acronyms, experimental definitions, and baseline comparisons (addressing WAVK).

We believe that establishing such a rigorous empirical foundation is essential for the field and align with scientific best practices, and we hope our work can serve as a valuable guide for future research.

Best,
The Authors of Submission 15785

---

### Note · Authors · 2026-01-05

I have read and agree with the venue's withdrawal policy on behalf of myself and my co-authors.